# BrainODE: Neural Shape Dynamics for Age- and Disease-aware Brain Trajectories

**Wonjung Park**[*]
KAIST
fabiola@kaist.ac.kr

**Suhyun Ahn**[*]
KAIST
ahn.ssu@kaist.ac.kr

**Maria C. Valdes Hernandez**
The University of Edinburgh
m.valdes-hernan@ed.ac.uk

**Susana Muñoz Maniega**
The University of Edinburgh
s.m.maniega@ed.ac.uk

**Jinah Park**
KAIST
jinahpark@kaist.ac.kr

## Abstract

We present `BrainODE`, a neural ordinary differential equation (ODE)-based framework for modeling continuous longitudinal deformations of brain shapes. `BrainODE` learns a deformation space over anatomically meaningful brain regions to facilitate early prediction of neurodegenerative disease progression. Addressing inherent challenges of longitudinal neuroimaging data—such as limited sample sizes, irregular temporal sampling, and substantial inter-subject variability—we propose a conditional neural ODE architecture that models shape dynamics with subject-specific age and cognitive status. To enable autoregressive forecasting of brain morphology from a single observation, we propose a pseudo-cognitive status embedding that allows progressive shape prediction across intermediate time points with predicted cognitive decline. Experiments show that `BrainODE` outperforms time-aware baselines in predicting future brain shapes, demonstrating strong generalization across longitudinal datasets with both regular and irregular time intervals.

## 1 Introduction

Precise modeling of disease progression in neurodegenerative disorders, particularly Alzheimer's disease (AD), is imperative for enabling early diagnosis and effective intervention [2, 28, 29]. The progressive and heterogeneous nature of these disorders poses significant challenges for longitudinal modeling [4, 22]. Besides, many studies have shown that the hippocampus, a key brain region implicated in AD, plays a critical role in accurate diagnosis and monitoring of disease trajectories. These highlight the potential of longitudinal models to support emerging therapeutic advancements toward targeted and personalized strategies [19]. Consequently, developing robust longitudinal modeling approaches has become important in improving clinical outcomes in AD.

To model neurodegenerative progression, directly learning from 3D brain MRI data is an intuitive approach. However, this strategy is often impractical due to the severe scarcity of longitudinal medical imaging data [5] and the inherently high dimensionality of MRI scans. Recent studies on the biological trustworthiness of generative models for brain imaging report that these limitations hinder the ability to capture complex brain changes driven by both aging and disease processes [1]. Furthermore, the hippocampus—a key brain region for diagnosing cognitive status—has only 1.7% of the full brain MRI volume. Although autoencoder architectures have been used to manage high-dimensional images in latent spaces to alleviate unaffordable GPU memory consumption, auto-encoded inputs are

---

[*]equal contribution

never exactly reconstructed after being decoded [14]. This poses a significant difficulty to deal with full brain MRI as cognitive status is often reflected in subtle structural changes within specific brain regions [16] such as the hippocampus. Therefore, efficient yet expressive representations with high precision such as surface meshes are required to address the challenge.

Beyond representations, modeling clinically plausible disease progression requires addressing several components of longitudinal medical data: 1) irregular time intervals, 2) medical priors, and 3) practical usage. First, longitudinal medical data consist of irregular time points, requiring methods that handle variable time intervals (*i.e.*, conventional inductive bias for uniform interval times is not suitable). Moreover, these methods should support updating predictions based on sequential longitudinal data. Second, driving clinically meaningful results can be achieved by incorporating demographic information, cognitive status, and morphological priors for deformations at scale [4, 23]. Especially, since neurodegenerative progression is gradual, not reversible, and not discrete, the method should be able to condition intermediate representations for the cognitive status $c_t \in [0, 1]$ spanning normal to AD. Lastly, given the constrained acquisition environment, the method should predict progression from a single time point, enhancing practical applicability. In addition, this capability is critical for overcoming the limitation of requiring multiple data points for interpolation at intermediate time points and extrapolation to future time points.

In this paper, we present a pioneering approach, BrainODE $f_\theta(V_t, c_t, t)$, which progressively models brain dynamics using neural ordinary differential equations (neural ODEs), simultaneously addressing the key components of longitudinal modeling, including irregular time points, medical priors, and practical usage. BrainODE learns the deformation space (*i.e.*, morphological prior) in brain shapes on longitudinal data of irregular time intervals under supervision of an ODE solver, enabling efficient modeling in shape spaces with the constant memory cost of neural ODEs [7]. For clinically plausible modeling, demographic information $t := age$ and cognitive status $c_t$ of subjects are integrated to condition our model. Contrary to using discrete or future cognitive status in previous studies, BrainODE is capable of embedding the continuous states in modeling hidden states $h(t) = f_\theta(V_t, c_t, t)$, consistent with the gradual and continuous nature of neurodegenerative progression.

Beyond plausible conditioning, we further propose the pseudo-cognitive status embedding method, which integrates pseudo-cognitive status shape sampling and a cognition estimator. Especially, by leveraging the knowledge of the gradual conversion and *condition-injectivity* of BrainODE, the sampling generates intermediate shapes $\tilde{V}_t$ and the corresponding status $\tilde{c}_t$ through interpolation between the forward and the backward trajectory of the converted subject whose diagnosis is changed from normal cognition to AD. Afterward, we employ a cognition estimator $\tau_\theta$, trained on pseudo-data and observation $(\tilde{V}_t, \tilde{c}_t, V_t, c_t)$, to provide smooth conditions for enhancing the fidelity of cognitive status for BrainODE. Moreover, through this estimator $\tau_\theta$, our novel method has a unique property that enhances the practical feasibility of clinically progressive modeling. Since the estimator predicts the severity of cognition from the geometry of the brain shape, BrainODE can seamlessly incorporate these predictions to model longitudinal progression.

In summary, our main contributions are as follows.

1. We propose a neural ODE-based brain shape deformation model specifically designed to address key challenges of longitudinal medical data, including irregular intervals, medical priors, and practical applicability.

2. Through extensive experiments comparing various methods on datasets with both regular and irregular time intervals, we validate and demonstrate the effectiveness of BrainODE in modeling brain shape dynamics.

3. We show that the pseudo-cognitive status embedding strengthens the BrainODE's applicability in longitudinal medical domains, enabling a robust and stable modeling for BrainODE in patients transitioning to neurodegenerative disorders, especially AD.

## 2   Related Work

To model longitudinal progression in neurodegenerative diseases, two primary approaches have been widely adopted: generative models and deformation-based methods. BrainODE emerges as a novel methodology that bridges the gap between deformation-based approaches, particularly ODE-based methods and flow models, offering a unique framework for learning brain dynamics.

**Generative models.** Building upon advancements in generative model, diffusion model-based methods have been proposed, embedding the aging prior in longitudinal progress modeling. For instance, BrLP [24] learns the Brain MRIs distribution and spatial-temporal consistency in the latent space of a VAE, covering various cognitive statuses of AD. However, its conditioned statuses are discretized and reflect future states, thus making it challenging to model continuous disease progression with the current status. Additionally, LoCI-DiffCom [35], another longitudinal generation method, models healthy infant aging on brain MRI scans. However, these image-based generative approaches [15, 24, 33] have common limitations: substantial computational costs, inconsistencies in subject identity due to Gaussian sampling (*i.e.*, subject-specificity is lost due to stochasticity in generation), and coarse resolution that fails to capture subtle yet clinically important changes. Furthermore, none of these methods is capable of modeling the clinical continuum of the AD trajectory as they rely on discrete condition values for the disease labeling. ConDOR [8], a recent study on longitudinal neurodegenerative diseases, generates volume values of brain regions using a conditional diffusion model. It employs ordinal regression for pseudo-intermediate sampling across longitudinal observations. While ConDOR addresses irregular intervals and predicts future values from a single point input, it produces less representative scalar values than volume grids or shapes and also relies on discrete ordinal categories for disease statuses.

**Neural ordinary differential equations.** Neural ODEs define the dynamics of hidden states by parameterizing their temporal derivatives with a neural network, enabling continuous-time modeling of complex dynamics [7, 27]. To handle irregular time intervals, Latent ODEs [27] employ variational autoencoders (VAEs) to model the likelihood of observations. However, autoencoding methods yield imperfect preservation due to the information loss [14] and are hard to train in modeling prior distribution when the data is scarce. For high-dimensional data, NODEO [31] is designed for 2D image registration tasks by learning continuous deformation fields between moving and fixed images.

In the medical image domain, LaTiM [34] performs NeuralODE-based classification tasks for diabetic retinopathy in 2D fundus images, predicting severity grades. For 3D brain MRIs, Lachinov *et. al.* [17] leverage a NeuralODE to model longitudinal progression on optical coherence tomography and brain MRI volumes for a single disease. Since directly modeling in voxel spaces requires huge computation costs, they project 3D feature representations into 2D space to solve an ODE solver. NODER [3] aims to predict missing timepoints in brain image sequences, requiring at least two prior timepoints as input to regress deformation fields. However, registration-based methods such as NODEO and NODER limits their practical applicability for predicting future shapes from a single observation. Moreover, it requires separate training for each individual brain image sequence, making it unsuitable for learning generalizable longitudinal dynamics. Furthermore, none of these approaches model the progressive transition from healthy to diseased states across continuous trajectories, and their reliance on image grids rather than volumetric shape spaces distinguishes them from our work.

**Flow models.** Continuous normalizing flow models have been recently proposed [11, 21], employing a differential ODE solver to generate bijective mappings through learned fluid dynamics. For instance, PointFlow [32] maps point cloud geometries to a captured prior distribution with a VAE, often resulting in loss of geometric details. Another representative work, ShapeFlow [14], directly learns the deformation function between geometries, preserving shape details. In contrast to our BrainODE, these flow-based methods assume uniform time intervals for one-to-one mappings between shapes and are thus not readily suited for irregularly sampled time series. ImageFlowNet [18] models subject-specific spatiotemporal dynamics in 2D medical images using neural ODEs/SDEs with multi-scale representations based on a U-Net architecture. The use of multi-scale representations in a joint space of neural ODEs/SDEs and pixel-level reconstruction results in substantial memory consumption, making it infeasible in 3D voxel space (e.g., over 80 GB for volumes of size $128^3$). In contrast, BrainODE employs mesh representations and PCA-based 3D shape parameterization, which significantly reduces computational requirements ($\sim$ 375 MB for training) while achieving high-precision shape modeling. In medical images, TimeFlow [13] learns a deformation function for registration in longitudinal brain MRIs with varying time intervals. However, generating transformations requires two time points of MRI scans for interpolation and extrapolation, limiting its applicability compared to BrainODE, which can predict disease progression from a single time point. Multi-Marginal Flow Matching (MMFM) [25] shares several core similarities with our BrainODE. Both methods employ a single shared model to learn across diverse conditions, support continuous dynamics modeling, handle irregular and multi-time-point data, and incorporate techniques for imputing mission observation.

However, MMFM is designed to model distributional translations, making it well-suited for capturing general patterns and group-level dynamics in low-dimensional scalar spaces. Consequently, it does not estimate explicit individual trajectories nor incorporate structural parameters that reflect anatomical geometry. In contrast, BrainODE embeds brain morphology directly into a parameter space and models continuous shape trajectories at both individual and disease group levels. This enables precise tracking and predicting of morphological changes for individual subject.

# 3 Method

## 3.1 BrainODE

**Definition of brain shapes.** Our BrainODE learns a continuous deformation trajectory of brain shapes from longitudinal datasets. We denote the longitudinal brain shape sequence for each subject as $V = \{V_{t_0}, V_{t_1}, \ldots, V_{t_N}\}$, where $N \geq 2$. Here, $t_i$ are irregular time points of the age at medical imaging acquisition. Each shape is reconstructed as a triangular mesh $X_i = \{V_i, \mathcal{F}\}$, where $V_i = \{v_1, v_2, \ldots, v_n\}$ denotes the ordered set of mesh vertices, and $\mathcal{F} = \{f_1, f_2, \ldots, f_m\}$ is the shared face connectivity across all shapes. Details on mesh reconstruction from raw MRI scans are provided in Appendix A.1.

To model brain shape deformation efficiently given scarce longitudinal datasets, we further reduce the dimensionality of the vertex space using principal component analysis (PCA), a widely used technique in shape modeling [6, 26]. Our approach is motivated by the following factors: (i) the necessity of preserving key shape features of blocky and noisy mesh surfaces derived from 1 mm$^3$ resolution MRIs, and (ii) the observation that local shape features do not change abruptly despite overall volume reduction caused by brain cell loss. Each brain shape is projected onto the PCA basis as $V_i \approx \sum_{j=1}^{k} \lambda_j^i e_j$ where $\Lambda_i = \{\lambda_1^i, \lambda_2^i, \ldots\}$ and $\mathcal{E} = \{e_1, e_2, \ldots\}$ are the set of PCA coefficients and eigenvectors, respectively, and $k$ is the number of retained components. To ensure accurate reconstruction from the reduced space, we empirically determined the optimal dimension $k = 150$ based on the explained variance ratio and reconstruction loss analysis (see Appendix A.2). Finally, the shape representations to train the longitudinal trajectory of brainODE is $\Lambda = \{\Lambda_{t_0}, \Lambda_{t_1}, \ldots, \Lambda_{t_N}\}$.

**Training deformation space of brain shapes.** BrainODE $f_\theta(\Lambda_t, c_t, t)$ models shape deformation as a function of the individual current shape $\Lambda_t$, cognitive status $c_t \in [0, 1]$, and normalized age $t \in [0, 1]$, defined as:

$$f_\theta(\Lambda_t, c_t, t) = \frac{d\Lambda_t(c_t, t)}{dt}. \tag{1}$$

Under the NeuralODE framework, brainODE estimates the shapes $\Lambda_{t+\Delta t}$ at the timepoint $t + \Delta t$ from the timepoint $t$ by solving the following formulation:

$$\Lambda_{t+\Delta t} = \Lambda_t + \int_t^{t+\Delta t} f_\theta(\Lambda_t, c_t, t) dt. \tag{2}$$

By using an ODE solver, BrainODE can model longitudinal data of irregular time intervals. It also incorporates the prior knowledge of brain atrophy and ventricle enlargement by leveraging cognitive status $c_t$ and age $t$. For a pair of longitudinal deformation from time $t_i$ to $t_j$, BrainODE is trained to seek a mapping $\Phi_\theta^{t_i t_j} : \mathbb{R}^k \mapsto \mathbb{R}^k$ that minimizes the distance between predicted deformed shapes $\Phi_\theta^{t_i t_j}(\Lambda_i, c_i)$ and the target shape $(\Lambda_j)$. The model is supervised using an L2 loss:

$$\mathcal{L}_2 = \left\| \Phi_\theta^{t_i t_j}(\Lambda_i, c_i) - \Lambda_j \right\|_2^2. \tag{3}$$

**Combinatorial training samples.** To fully utilize the longitudinal datasets with arbitrary $N$ observations for alleviating the data deficiency, we design the training process for BrainODE to use the entire combinations of time points. Furthermore, BrainODE learns not only a forward trajectory of neurodegenerative disease progression but also the backward trajectory to lead the invertible shape deformation (i.e., $\Phi_\theta^{t_j t_i}(\Phi_\theta^{t_i t_j}(\Lambda_i, c_i), c_i) = \Lambda_i$). Algorithm 1 elaborates the training process for BrainODE.

---

**Algorithm 1** BrainODE training process

---

1: **Input:**
2: A set of observed shapes $\Lambda = [\Lambda_0, \Lambda_1, \ldots, \Lambda_{N-1}]$,
3: corresponding cognitive status $c = [c_0, c_1, \ldots, c_{N-1}]$, age $t = [t_0, t_1, \ldots, t_{N-1}]$
4: $N \leftarrow \text{length}(\Lambda)$, loss $\leftarrow 0$
$\qquad\qquad\qquad\qquad\qquad\qquad\qquad\qquad\qquad\qquad\qquad\qquad\qquad$ ▷ Forward trajectory
5: **for** $i = 0$ to $N - 2$ **do**
6: $\quad \Lambda_{\text{pred}}[i:N] \leftarrow \texttt{ODEintegral}(\text{input} = (\Lambda[i], c[i], t[i]), \text{target\_time} = t[i:N])$ $\quad$ ▷ Eq. (2)
7: $\quad \text{loss} \leftarrow \text{loss} + \mathcal{L}_2(\Lambda_{\text{pred}}[i:N], \Lambda[i:N])$
8: **end for**
$\qquad\qquad\qquad\qquad\qquad\qquad\qquad\qquad\qquad\qquad\qquad\qquad\qquad$ ▷ Backward trajectory
9: **for** $i = 1$ to $N - 1$ **do**
10: $\quad \Lambda_{\text{pred}}[0:i] \leftarrow \texttt{ODEintegral}(\text{input} = (\Lambda[i], c[i], t[i]), \text{target\_time} = t[0:i])$ $\quad$ ▷ Eq.(2)
11: $\quad \text{loss} \leftarrow \text{loss} + \mathcal{L}_2(\Lambda_{\text{pred}}[0:i], \Lambda[0:i])$
12: **end for**

---

## 3.2 Pseudo-cognitive status embedding

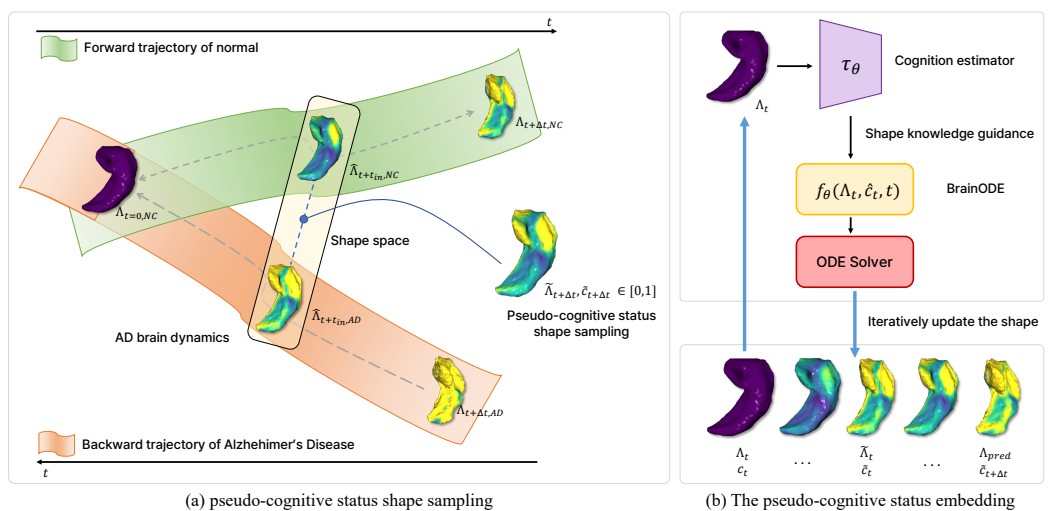

(a) pseudo-cognitive status shape sampling $\qquad\qquad$ (b) The pseudo-cognitive status embedding

Figure 1: Illustration of the proposed methods of BrainODE. (a) The framework of the pseudo-cognitive status shape sampling for continuous progression modeling of neurodegenerative disorders. (b) The pseudo-cognitive status embedding of BrainODE is guided by a cognition estimator $\tau_\theta$.

**Pseudo-cognitive status shape sampling.** Beyond a longitudinal prediction for the unchanged cognitive status, to achieve a clinically useful brain shape progression for early diagnosis, it is essential to reflect the gradual decline in cognitive status in modeling. However, discrete diagnostic labels such as normal and disease as well as the severe scarcity of converted cases make it difficult to model the conversion toward disease. To address this, we introduce a pseudo-cognitive status shape sampling, bridging the absence of explicitly intermediate cognition data. Using *condition-injectivity* and both trajectories of BrainODE in subsection 3.1, we sample intermediate brain shapes along their shape trajectory $\Lambda_t \mapsto \Lambda_{t+\Delta t}$ by assuming a smooth morphological transition over time as the prior knowledge of progressive brain atrophy by aging and disease. The intermediate shape at time $t + t_{\text{in}}$ (where $0 \leq t_{\text{in}} \leq \Delta t$) is interpolated between predicted shapes from the forward trajectory of $\Lambda_t$ with $c_t = 0$ and the backward trajectory of $\Lambda_{t+\Delta t}$ with $c_{t+\Delta t} = 1$, using the learned deformation $\Phi_\theta$ of BrainODE. The pseudo-cognitive status $\tilde{c} \in [0, 1]$ is defined as the relative position between the two time points. The final formulation for the pseudo-cognitive status and its shape is:

$$\tilde{\Lambda}_{t+t_{\text{in}}} = \frac{1}{2}\left(\Phi_\theta^{t, t+t_{\text{in}}}(\Lambda_t, c_t = 0) + \Phi_\theta^{t+\Delta t, t+t_{\text{in}}}(\Lambda_{t+\Delta t}, c_{t+\Delta t} = 1)\right), \quad \tilde{c}_{t+t_{\text{in}}} = \frac{t_{\text{in}}}{\Delta t} \qquad (4)$$

These pseudo-labels and their corresponding interpolated shapes are used to train BrainODE to learn deformation dynamics that span the cognitive continuum.

**Progressive longitudinal shape prediction with cognition estimator.** To predict future brain shapes and their associated cognitive status, we introduce a progressive estimation framework for longitudinal shape progression and the corresponding cognitive status. Prior work demonstrated that hippocampal binary masks can serve as effective biomarkers for distinguishing between NC and AD, achieving over 90% accuracy [19]. Inspired by this insight, we developed a cognition estimator $\tau_\theta$ trained on an augmented dataset composed of $(\Lambda_t, c_t, \tilde{\Lambda}_t, \tilde{c}_t)$. Through this training, the cognition estimator learns longitudinal shape changes corresponding to continuous cognitive status.

During inference, using the pseudo-cognitive status shape sampling and cognition estimator, we update the future prediction progressively using the estimated intermediate shapes and cognitive status (*i.e.*, risk probability of AD). We term this progressive modeling as the pseudo-cognitive status embedding, depicted in Algorithm 2.

---

**Algorithm 2** The pseudo-cognitive status embedding

---

**Require:** $\Lambda(t)$: Current brain shape at time $t$,    $N$: Number of intermediate steps
1:    $\Delta t$: Target time interval,    $\tau_\theta$: Moel of cognition estimator
2:
3: Initialize $\Lambda_{\text{mid\_pred}} \leftarrow \Lambda(t)$
4: **for** $i = 0$ to $N - 1$ **do**
5:    $\hat{c} \leftarrow \tau_\theta(\Lambda_{\text{mid\_pred}})$
6:    $\Lambda_{\text{mid\_pred}} \leftarrow \texttt{ODEIntegral}(\text{input} = (\Lambda_{\text{mid\_pred}}, \hat{c}, t + i \cdot \frac{\Delta t}{N}), \text{target\_time} = t + (i+1) \cdot \frac{\Delta t}{N})$
7: **end for**                                                                                           ▷ Eq. (2)
8: $\Lambda_{\text{pred}} \leftarrow \Lambda_{\text{mid\_pred}}$
9: **return** $\Lambda_{\text{pred}}$

---

## 4   Experiments

In this section, we demonstrate the comparison of BrainODE with baseline methods for time series and then discuss the effect of our proposed components. We conduct experiments using four different longitudinal brain MRI datasets of normal cognition and Alzheimer's disease (AD), comprising both regular and irregular time intervals: for regular intervals, the Lothian Birth Cohorts 1936 (LBC1936) [9] and the Australian Imaging, Biomarker and Lifestyle (AIBL) datasets [10], each collected at fixed 3 and 1.5 year intervals, respectively; for irregular intervals, the Alzheimer's Disease Neuroimaging Initiative (ADNI) [12] and the Open Access Series of Imaging Studies (OASIS) datasets [20], with varying acquisition intervals reflecting real-world clinical settings. We provide detailed implementations and training settings of BrainODE in Appendix A. Also, the demographics and cognitive statuses of these datasets and preprocessing methods for shape representations are detailed in Appendix B.

**Baselines.** To provide a comprehensive evaluation, we employ baseline models that can jointly model shapes and temporal sequences, including traditional time-aware architectures, a flow-based model, a neural ODE-based model, and a generative model. As conventional time-aware baselines, we include recurrent neural networks (RNNs), an RNN variant for irregular intervals (RNN-Decay), and long short-term memory networks (LSTMs). For flow-based modeling, we adopt ShapeFlow [14], which learns a continuous deformation field between pairs of geometric shapes. In addition, we employ Latent ODE [27], a neural ODE-based model that can handle irregular time intervals by integrating an ODE solver parameterized by a neural network. We also employed a simple yet plausible baseline, linear extrapolation for subjects with two or more available observations. For image generative models, we adopt BrLP [24], which generates longitudinally predicted MRIs conditioned by discrete cognitive statuses and volume of brain regions. Regarding image registration-based approaches, we also report the evaluation results of NODER [3] in Appendix C.6, as it cannot perform prediction from a single time point and is therefore not included in the main comparison.

**Evaluation.** Given the varying number of longitudinal observations $N$ ranging from 2 to 5 across the datasets, we formulate the brain disease progression modeling into two tasks: (1) predicting the latest shapes for each subject using four observed time points (4-shot prediction), and (2) predicting

| Method | LBC [9] | | | | AIBL [10] | | | |
|---|---|---|---|---|---|---|---|---|
| | 4-shot | | 1-shot | | 4-shot | | 1-shot | |
| | hippo | LV | hippo | LV | hippo | LV | hippo | LV |
| Linear extrap. | 0.737 | 2.075 | - | - | 0.666 | 2.013 | - | - |
| RNN | 1.060 | 5.528 | 12.801 | 50.829 | 0.964 | 5.561 | 10.914 | 52.418 |
| LSTM | 1.044 | 5.723 | 9.978 | 37.948 | 0.957 | 6.745 | 7.848 | 39.572 |
| RNN-Decay | 1.075 | 5.549 | 0.787 | 5.703 | 0.967 | 5.587 | 0.780 | 5.601 |
| ShapeFlow [14] | 0.652 | 7.112 | 0.576 | 4.939 | 0.776 | 3.149 | 0.567 | 2.796 |
| LatentODE [27] | 0.880 | 5.759 | 0.895 | 6.663 | 1.126 | 8.095 | 1.079 | 6.800 |
| BrLP [24] | 1.078 | 2.230 | 1.054 | 2.067 | 1.019 | 1.893 | 1.031 | 2.042 |
| Ours | **0.488** | **1.630** | **0.365** | **1.743** | **0.461** | **1.635** | **0.406** | **1.708** |

Table 1: Quantitative evaluation of shape prediction performance in Euclidean distance on the LBC1936 and AIBL test sets with regularly sampled time points.

| Method | 4-shot | | 1-shot | |
|---|---|---|---|---|
| | hippo | LV | hippo | LV |
| Linear extrap. | 0.898 | 2.240 | - | - |
| RNN-Decay | 0.984 | 6.444 | 0.832 | 6.306 |
| LatentODE [27] | 1.215 | 5.300 | 1.119 | 4.463 |
| BrLP [24] | 1.017 | 1.962 | 1.031 | 2.085 |
| Ours | **0.543** | **1.959** | **0.492** | **1.673** |

Table 2: Quantitative evaluation of shape prediction performance in Euclidean distance on ADNI, OASIS, AIBL test sets with irregular time points.

| | Shape acc | | $\tau_\theta$ acc |
|---|---|---|---|
| | NC & AD | CONV | |
| Baseline | 0.636 | 0.282 | 0.883 |
| + $\tau_\theta$ only | 0.611 | 0.310 | 0.883 |
| + pseduo only | 0.619 | **0.178** | 0.875 |
| BrainODE | **0.606** | 0.216 | **0.891** |

Table 3: Ablation results of BrainODE. We report shape prediction performance (mm) and diagnosis accuracy by the cognition estimator. CONV denotes the converted cases.

the latest shapes from a single observation point (1-shot prediction). These tasks are intended to reflect the practical characteristics of medical longitudinal datasets as well as to evaluate whether the baseline methods and BrainODE can address the crucial components for clinically plausible disease progression modeling discussed in section 1. The lateral ventricles (LV) and hippocampus (hippo), key brain regions for AD diagnosis, are selected as representative target shapes. To assess the shape modeling quality, we measure the average Euclidean distance between the predicted and ground truth shapes. For image-based prediction with BrLP [24], Chamfer distances on the boundary points of the segmented brain regions are measured. For pair comparisons, all methods are trained with a single type of time interval between the regular and irregular datasets.

For pair comparisons, all models are trained and evaluated separately for the two settings, and the regular and irregular datasets (*i.e.*, the subjects for each task are different patients).

## 4.1 Experimental Results

**Regular time interval predictions.** We first discuss prediction results on datasets with regular sampling intervals, LBC1936 and AIBL. As their intervals differ, we report the results separately.

Across all settings, BrainODE consistently outperforms baseline methods, as shown in Table 1. Especially, in the 1-shot tasks, our model demonstrates a clear advantage in capturing both subtle and large-scale morphological changes of the brain progression. Although RNNs and LSTMs benefit from inductive biases under regular intervals, they exhibit limited capability in modeling neurodegenerative trajectories, especially in the 1-shot setting, due to their inability to capture continuous temporal dynamics effectively. Among deep learning-based methods, RNN-Decay and BrLP achieve competitive results with BrainODE. However, on the LBC1936 dataset, their performance even falls below that of simple linear extrapolation, underscoring the inherent difficulty of modeling shape trajectories in progressive neurodegeneration. ShapeFlow shows notably inferior performance in LV modeling on LBC1936, which has longer temporal intervals than AIBL, indicating its limitations in capturing the brain dynamics driven by aging. This highlights the importance of methods that incorporate both effective representation and domain-specific knowledge.

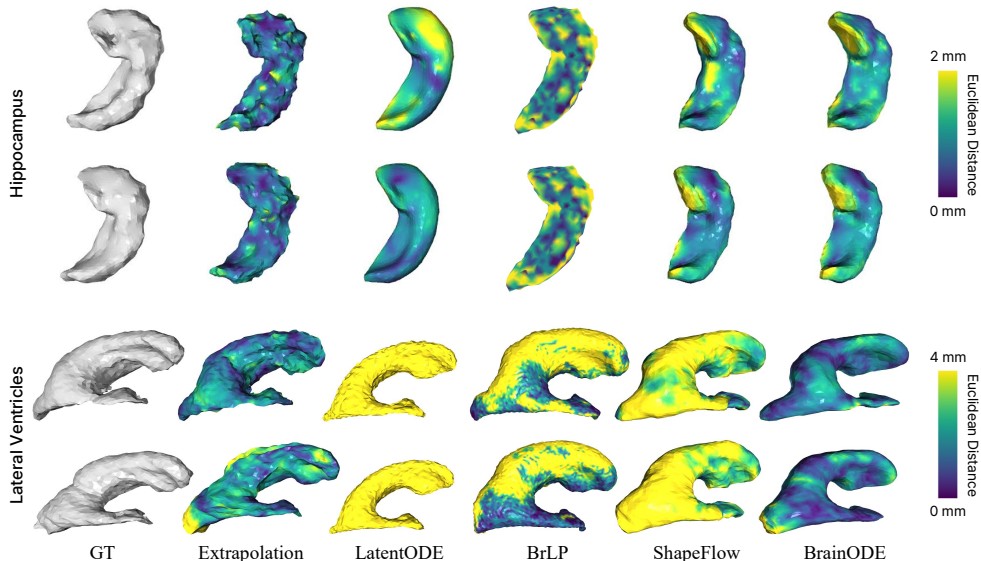

Figure 2: Qualitative results in modeling the LV and hippocampus shapes.

Interestingly, BrLP exhibits a larger discrepancy from BrainODE in the hippocampus than in the LV. A plausible interpretation is that the hippocampus—occupying a much smaller proportion of the brain compared to the LV—poses challenges for voxel-based autoencoders in capturing its fine-grained morphology. LatentODE also demonstrates inferior performance since it relies on an autoencoder to obtain latent features of mesh representations. In contrast, BrainODE preserves structural fidelity in such subregions by leveraging its deformation space modeling method.

**Irregular time interval predictions.** We further evaluate BrainODE on a unified dataset combining ADNI, OASIS, and AIBL, which contain irregular sampling intervals. As described in Table 2, BrainODE again achieves superior performance compared to other methods capable of handling irregular time intervals. However, unlike the previous experiment with regular intervals, BrainODE's 4-shot prediction results exhibit degraded performance relative to Table 1. This decline can be attributed to the increased complexity of irregular datasets; for instance, ADNI and OASIS include subjects with a wide age range (*e.g.*, 55–93 years), introducing greater inter-subject variability in disease progression. In contrast, LBC1936 consists of subjects with identical ages across each 3-year acquisition, and AIBL features shorter intervals of approximately 1.5 years, resulting in more temporally consistent trajectories.

In comparison, BrLP shows performance improvement in the irregular 4-shot setting. However, compared to deformation methods, we found that this image-generative approach cannot preserve the subject consistency due to autoencoding and prior sampling. While iterative sampling of BrLP mitigates this limitation statistically, it cannot guarantee that the generated subject identity matches the input, as further discussed in subsection C.5.

### 4.2 Qualitative results and condition fidelity

**Qualitative results.** For a qualitative comparison, we visualize the Euclidean distance error maps on the predicted meshes in Figure 2, illustrating the performance of 4-shot predictions using linear extrapolation, LatentODE, BrLP, ShapeFlow, and our BrainODE. Consistent with the quantitative results in Table 1, BrainODE exhibits the smallest Euclidean distance errors for both the LV and hippocampus shapes from the ground truth (GT), with error maps primarily in the blue range. Linear extrapolation yields plausible shapes but requires more than two time points and generates noisier surfaces, as indicated by the yellow and green regions (up to 2 mm errors). Furthermore, when the intral becomes greater, this method produces abnormal shapes and therefore is not appropriate for shape prediction (see Figure 14). BrLP predicts longitudinal individual hippocampal and LV shapes with spatial-temporal consistency of each subject, but its voxel-based generation introduces significant deviations from the GT, with errors reaching up to 4 mm (yellow areas). LatentODE struggles to

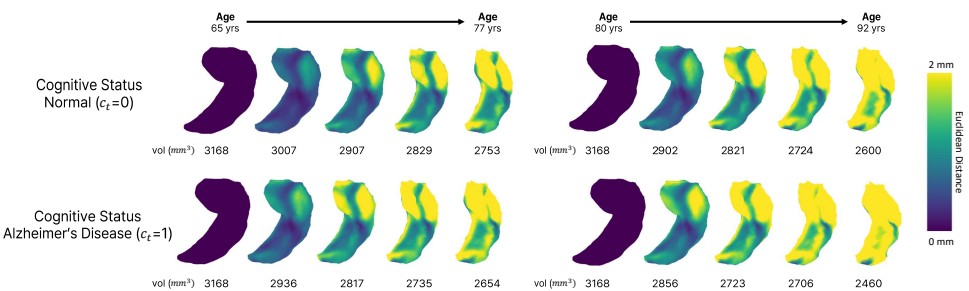

Figure 3: Demonstration of shape prediction by BrainODE with the age and cognitive status conditions. The hippocampal shapes in the first row are from the age of 65 to 77 with 3-year intervals. The second column is shapes from the age of 80 to 92.

capture unique shape features from individuals, often producing nearly identical LV shapes for different subjects, as evidenced by uniform error distributions. ShapeFlow inherits individual shape characteristics via its flow-based ODE, but its assumption of uniform time intervals during training leads to higher errors (2–4 mm) compared to BrainODE, which effectively adapts to varying intervals. Overall, BrainODE demonstrates superior performance with smoother error distributions that closely align with the GT shapes across both regions.

**Condition-injective shape modeling.** To evaluate the qualitative effects of condition injection in BrainODE, we visualize predicted hippocampal meshes under varying injected conditions in Figure 3. Specifically, we compare predictions starting from the same initial hippocampal shape while varying age (65 and 80 years) and cognitive status ($c = 0$ for normal cognition, $c = 1$ for declined cognition). As shown, BrainODE models longitudinal hippocampal shape changes while reflecting medical priors that older age and declined cognition accelerate hippocampal volume degradation [4]. For a starting age of 65, a cognitively normal subject ($c = 0$) exhibits a hippocampal volume reduction from 3168 to 2753 $mm^3$, whereas a subject with declined cognitive status ($c = 1$) shows a larger reduction from 3168 to 2654 $mm^3$. Furthermore, for the same cognitive status of AD, older subjects starting at age 80 undergo a more significant volume reduction (*e.g.*, from 3168 to 2460 $mm^3$) compared to their younger counterparts. These results highlight the *condition-injectivity* of BrainODE to predict deformation based on given conditions. Further quantitative analysis of condition-injectivity is provided in subsection C.4.

### 4.3 Evaluation on the Pseudo-cognitive Status Embedding

We perform an ablation study on BrainODE and its pseudo-cognitive status embedding, which includes the pseudo-cognitive status shape sampling and the cognition estimator $\tau_\theta$. Beyond shape modeling, we conduct a diagnosis task for neurodegenerative disorders using $\tau_\theta$ and report the classification accuracy. Table 3 summarizes the results, where the vanilla Baseline (described in subsection 3.1) excludes both the pseudo-cognitive status shape sampling and the cognition estimator. The pretrained $\tau_\theta$ shows an accuracy of 88.3% in diagnosing normal cognition (NC) and AD based on shape representations. Compared to the Baseline, incorporating only the cognitive estimator $\tau_\theta$ in the BrainODE pipeline improves shape modeling accuracy for NC and AD (from 0.636 to 0.611), but reduces performance for converted cases (CONV, from 0.282 to 0.310). In contrast, applying pseudo-cognitive status shape sampling to augment the dataset enhances shape modeling precision in terms of Euclidean distance, particularly for the CONV cases (from 0.282 to 0.178). However, it slightly decreases the diagnosis accuracy (from 0.883 to 0.875) compared to training with only observed data.

When both approaches are combined, BrainODE achieves the lowest shape modeling errors for NC and AD (0.606) and a competitive performance for CONV (0.216), outperforming the Baseline across all categories. This suggests that the estimated cognitive status $\tilde{c}_t$ provides geometric cues that enhance BrainODE's understanding of neurodegenerative disorders, enabling the reflection of this knowledge in the shape modeling process within the shape space. Notably, during inference, applying the pseudo-cognitive status embedding method boosts the prediction accuracy to 89.1% the highest among all configurations, highlighting its effectiveness in real-world scenarios.

# 5 Discussion

BrainODE is the first neural ODE-based framework for modeling longitudinal 3D brain shape trajectories. We focus on precise modeling of shape deformation in clinically important subregions, beyond the prior works that mainly targeted whole-brain image prediction.

Our contributions are threefold. First, BrainODE bridges neural ODE-based 3D shape modeling with longitudinal neurodegenerative disease. We propose a pipeline consisting of mesh reconstruction for longitudinal shape representation, PCA-based parameterization, and a unified deformation-based ODE framework across disease groups. While neural ODEs have shown promise in continuous-time modeling, they have not been applied to predict longitudinal brain subregion shapes. To fill this gap, we convert brain images to fine-grained meshes and represent them in a compact PCA space. Then, we introduce BrainODE with continuous conditioning, pseudo-status embedding, and iterative shape sampling, for the model stability and disease-continuum plausibility. Our model achieves precise subject-level trajectory prediction with low computational cost (~375MB for training), while maintaining fidelity at both the individual and group levels.

Second, BrainODE is the first large-scale application of longitudinal brain shape modeling. To the best of our knowledge, BrainODE is trained on multi-site datasets with 30 years age span. Through large-scale experiments, we demonstrate that BrainODE effectively captures brain morphological patterns and aging dynamics across both normal and AD trajectories. Furthermore, we identify fundamental limitations in existing generative image-based models, which fail to precisely capture subregion-level deformation, and we offer straightforward adaptations of ODE methods to address this limitation.

Third, we define and simultaneously address key challenges unique to longitudinal brain shape modeling. These include data scarcity, irregular sampling intervals, the integration of medical priors, and the ability to predict only from a single observation. By defining these considerations upfront, BrainODE provides a foundational blueprint for this brain morphology prediction area.

**Limitations and Future Work.** Despite these advances, BrainODE has several limitations. In this work, we utilize two cognitive status groups: normal cognition (NC) and Alzheimer's disease (AD). Although some datasets include mild cognitive impairment (MCI) as a separate category and intermediate cognition between NC and AD, we excluded MCI from BrainODE modeling due to its ambiguous clinical definition and variability in progression (*e.g.*, assigning $c := 0.5$ for MCI seems straightforward yet cannot reflect severity in the values). In future work, we aim to extend BrainODE to incorporate NC, MCI, and AD by establishing more robust and discriminative criteria for cognitive status. For subjects with multiple longitudinal observations ($n \geq 1$), BrainODE can generate $n$ predicted shapes corresponding to each time point. To evaluate $n$-shot prediction performance, we simply average these predicted shapes. Developing more advanced aggregation strategies—such as attention-based or time-aware mechanisms—remains an important direction for future research to better capture longitudinal progression. As a pioneering model for brain shape prediction under varying cognitive conditions, BrainODE introduces a pseudo-cognitive status embedding that enables intermediate shape prediction and the corresponding cognition estimation. In this work, we primarily estimate cognitive status using hippocampal shapes, guided by prior research [19, 30]. In future work, we plan to explore BrainODE and congnition estimator to incorporate additional brain regions.

# 6 Conclusion

We introduce BrainODE, a pioneering approach for predicting longitudinal brain shape deformations, taking into account individual demographics and cognitive status. To address the challenges of longitudinal prediction, including irregular time intervals, embedding medical priors, and practical usage, we propose BrainODE within a neural ODE-based architecture that smoothly deforms input shapes to target shapes at future time points. In addition, we introduce a pseudo-cognitive status embedding to progressively predict future shapes over intermediate time points with predicted cognitive decline using the proposed cognition estimator. Through both quantitative and qualitative experiments, we demonstrate that BrainODE effectively models brain dynamics over time, outperforming baseline methods. Given the importance of accurately predicting brain shape changes in key regions for disease diagnosis, our work focuses primarily on precise longitudinal brain shape modeling. Future work will extend this approach to other brain regions and to clinical applications of early diagnosis.

## Acknowledgement

This work was supported by the National Research Foundation of Korea(NRF) grant funded by the Korea government(MSIT) (RS-2024-00508681,Establishment of Korea-UK preclinical/clinical joint research center to develop diagnosis and treatment strategy for neurodegenerative diseases) and Institute for Information & communications Technology Promotion (IITP) grant funded by the Korea government(MSIT) (No.00223446, Development of object-oriented synthetic data generation and evaluation methods).

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

# A  Implementation details

This section provides an overview of the design choices and implementation details for BrainODE. We begin by detailing the longitudinal brain shape reconstruction process from 3D brain MRI scans, which involves segmentation and reconstruction of the hippocampal and lateral ventricle (LV) shapes. Subsequently, we evaluate the effectiveness of principal component analysis for representing variable brain shapes, reducing dimensionality while preserving key shape features. Afterward, we present details of the neural architecture of BrainODE and its training strategies. Our source code, including data preprocessing scripts and model configurations, is publicly available at `https://github.com/PWonjung/BrainODE`

## A.1  Longitudinal brain shape reconstruction

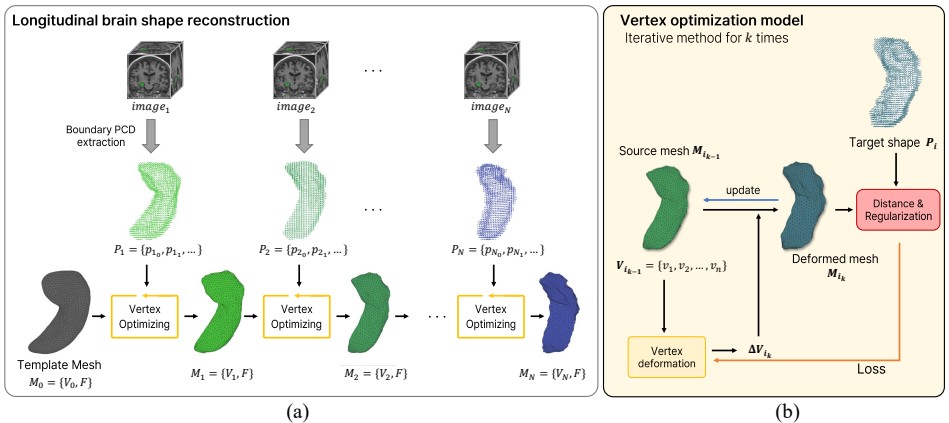

Figure 4: Illustration of brain shape reconstructions for a single subject by iteratively deforming a template mesh to match longitudinal MRI scans.

We reconstruct the longitudinal brain shapes $V = \{V_1, V_2, ..., V_n\}$ by modifying the brain shapes modeling method proposed by Park *et al.* [11]. Their approach reconstructs cross-sectional individual brain shapes by iteratively deforming a template mesh to align with target shapes, which are represented as point clouds extracted from the boundaries of segmented 3D brain MRI masks. To adapt this cross-sectional method for longitudinal analysis, we sequentially update a template mesh $M_0$ over time, using the deformed mesh from the previous time point, $M_{i-1}$, as the initial mesh for the next step, $M_i$. The process begins with the first observed MRI scan and proceeds iteratively through subsequent time points, optimizing vertex positions to minimize both distance and regularization loss, as illustrated in Figure 4.

As part of this pipeline, we use SynthSeg [1] to segment brain regions from MRI scans. From the boundaries of the segmented masks, point clouds $P_i$ representing the target shapes are constructed. The initial shape $V_1$ is obtained by deforming the template mesh vertices $V_0$ to fit the target point cloud $P_1$. Then, each subsequent shape $V_i$ is reconstructed by deforming the previous mesh $V_{i-1}$ to align with the corresponding target $P_i$, using a vertex optimization model.

The vertex optimization model (Figure 4(b)) follows the optimization strategy of Point2Mesh [4], which iteratively updates the vertices to align with the target point cloud. The objective function $\mathcal{L}$ consists of two components: 1) a distance loss $\mathcal{L}_{dist}$, which measures the discrepancy between the deformed mesh $V_i$ and the target point cloud $P_i$, and 2) a regularization loss $\mathcal{L}_{reg}$, which encourages smooth and plausible deformations.

Specifically, the distance loss $\mathcal{L}_{dist}$ combines the Chamfer distance $\mathcal{L}_{cf}$ between $V_i$ and $P_i$, and the point-to-face distance $\mathcal{L}_{pm}$ between $P_i$ and the mesh surface $M_i$ (*i.e.*, $\mathcal{L}_{dist} = \lambda_{cf}\mathcal{L}_{cf} + \lambda_{pm}\mathcal{L}_{pm}$). The regularization loss $\mathcal{L}_{reg}$ includes vertex displacement $||\Delta vert||_2$, normal displacement $||\Delta norm||_2$, edge length variance $\mathcal{L}_{edge}$ to prevent mesh distortion, normal consistency $\mathcal{L}_{cons}(norm)$, and Laplacian loss $\mathcal{L}_{lap}$ for surface smoothness. The final form of the loss function used in the vertex optimization model is defined as follows:

$$\mathcal{L} = \mathcal{L}_{dist} + \mathcal{L}_{reg} \tag{5}$$

where $\quad \mathcal{L}_{reg} = \lambda_{vert}||\Delta vert||_2 + \lambda_{norm}||\Delta norm||_2 + \lambda_{edge}\mathcal{L}_{edge} + \lambda_{cons}\mathcal{L}_{cons}(norm) + \lambda_{lap}\mathcal{L}_{lap}.$

We empirically set hyperparameters for each loss as $\{\lambda_{cf}, \lambda_{pm}, \lambda_{vert}, \lambda_{norm}, \lambda_{edge}, \lambda_{cons}, \lambda_{lap}\} = \{0.5, 3, 1, 1, 1500, 1, 5\}$.

## A.2 Effect of PCA on shape reconstruction

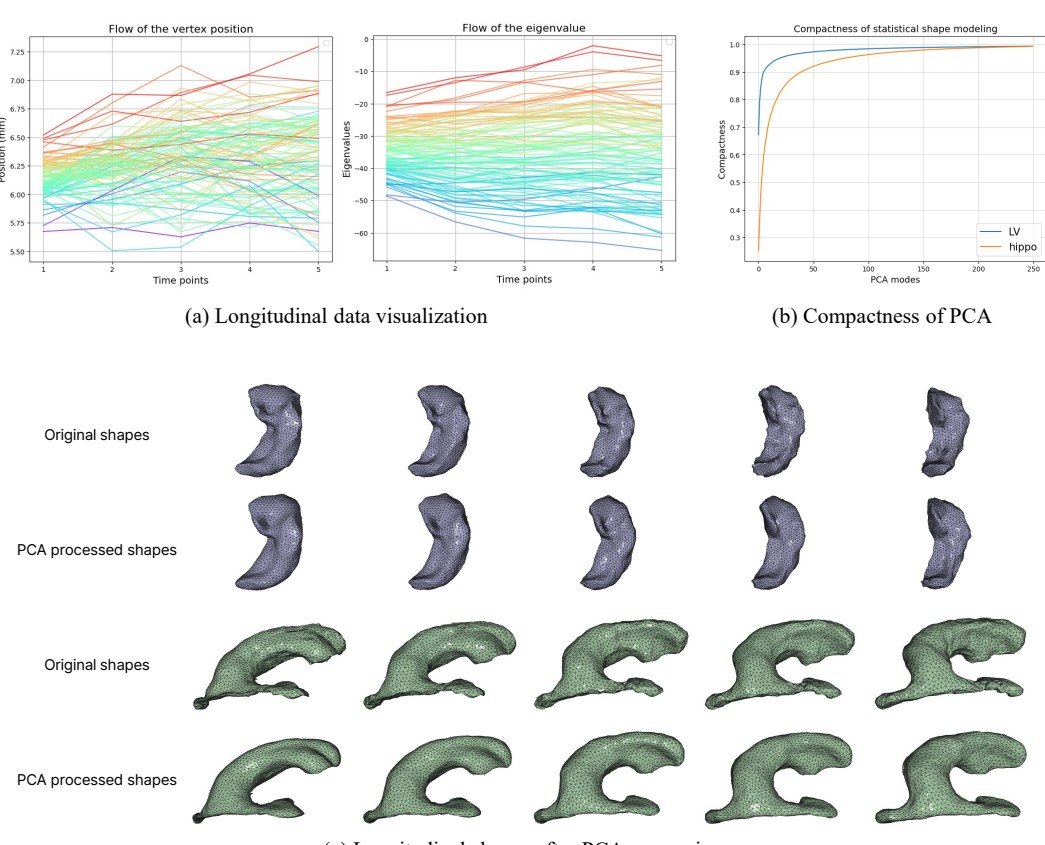

(a) Longitudinal data visualization

(b) Compactness of PCA

(c) Longitudinal shapes after PCA processing

Figure 5: Impacts of shape dimensionality reduction using PCA. (a) Visualization of vertex positions and the first PCA eigenvalue across longitudinal observations from the LBC1936 dataset. (b) Compactness measures of hippocampus and LV across varying numbers of PCA modes. (c) Qualitative comparison between original shapes and their PCA-reconstructed shapes with 150 modes.

**Motivation of PCA.** We adopt principal component analysis (PCA) to parameterize brain region shapes for longitudinal modeling, as described in subsection 3.1. Rather than using raw vertex-level inputs or latent codes from autoencoders, we choose PCA-based representations to facilitate reproducible and scalable research. This choice is inspired by prior work such as MANO [13], which models shape variation of hands using a low-dimensional parametric space. Building on this foundation, subsequent research has demonstrated the effectiveness of directly predicting such parameters for shape estimation tasks [5, 14, 16]. In a similar spirit, we leverage parametric modeling of brain shapes to construct compact and anatomically meaningful shape spaces to support the learning of the longitudinal shape dynamics.

Another key motivation for using PCA is its efficiency in shape representation. We visualize the vertex positions of brain shapes and the first eigenvalues across longitudinal time points in Figure 5(a). The raw vertex positions of hippocampal and LV shapes exhibit irregular deformations over time,

making it difficult to identify clear trends. This irregularity arises from the complex geometry of the original shape representation, which includes high-frequency artifacts (*e.g.*, bumpy surfaces due to image-level noise) and substantial inter-subject variability, as illustrated in Figure 5(c). In contrast, PCA-derived eigenvalues show more consistent trends across longitudinal time points, enabling stable progressive modeling of brain shape dynamics. Moreover, shapes reconstructed using PCA exhibit smoother surfaces by filtering out high-frequency noise while preserving essential geometric features (Figure 5(c)). This dimensionality reduction and smoothing effect of PCA not only simplifies the shape representation but also supports plausible modeling by providing stable inputs for capturing temporal progression in neurodegenerative disease.

**Quantitative results of PCA.** To quantitatively evaluate the effectiveness of PCA-based shape representation, we measured the compactness of the statistical shape models for both the hippocampus and LV, where Compactness$(k) = \frac{\sum_{i=1}^{k} \lambda_i}{\sum_{i=1}^{N} \lambda_i}$ for the first $k$ modes. As depicted in Figure 5(b), the compactness reaches 0.980 for the hippocampus and 0.989 for the LV with $k = 150$ modes, indicating that over 98% of the total shape variability is preserved in both cases.

| Shapes | PCA Modes | Dist. (mm) |
|--------|-----------|------------|
| $V$ | - | 0.2642 |
| | 50 | 0.3028 |
| | 100 | 0.3005 |
| $\Lambda$ | 150 | **0.2640** |
| | 200 | 0.2829 |
| | 300 | 0.2701 |

Table 4: Prediction loss in Euclidean distances (mm) when using brain shapes $V$ as-is and using PCA coefficients $\Lambda$ on hippocampal shapes of LBC1936.

In addition, to validate the use of PCA coefficients as shape representations, we compare BrainODE performance using original vertex-based shapes $V$ and PCA-projected shapes $\Lambda$ in modeling the hippocampus. As shown in Table 4, the use of PCA coefficients not only results in smoother reconstructed surfaces but also improves the prediction accuracy. The ablation study shows that using 150 components provides a favorable trade-off between reconstruction fidelity and dimensionality reduction.

## A.3 Neural architecture of BrainODE

**BrainODE.** We design the BrainODE function $f_\theta$ as a neural network with a self-attention mechanism to model condition-injective shape dynamics. The input to $f_\theta$ consists of a shape embedding $V \in \mathbb{R}^{150}$, a scalar time $t \in \mathbb{R}^1$, and a scalar cognitive status $c \in \mathbb{R}^1$. These are concatenated into a single vector and passed through three linear projections to form the query, key, and value. The attention output is then processed by fully connected layers with GeLU activation. The output $h(\cdot)$ is a 150-dimensional vector representing the velocity in the shape space at time $t$.

**Cognition estimator.** We adopt the cognition estimator $\tau_\theta$ as a simple 3D CNN architecture, followed by a prior study to diagnose Alzheimer's disease with the hippocampus masks [9]. The input to $\tau_\theta$ is a shape in voxel representation, and the output is the estimated cognitive status $c_t \in [0, 1]$.

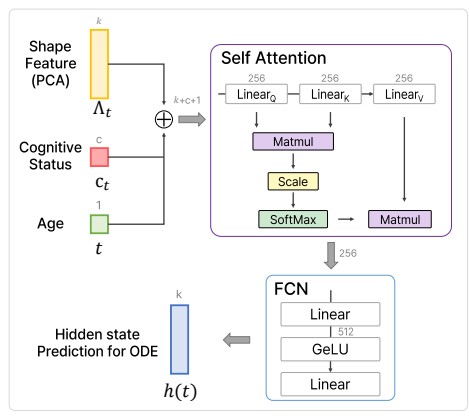

Figure 6: BrainODE architecture

## A.4 Training details of BrainODE

The input to BrainODE $f_\theta(\Lambda, t, c)$ consists of PCA-reduced shape coefficients $\Lambda$, normalized age $t$, and cognitive status $c$. Each brain shape $V_i = \{v_1, v_2, ..., v_n\}$ is represented as vertices of a triangular mesh, where the coordinates are in real-world brain sizes with units in mm. The typical spatial extents of the hippocampus and LV are approximately (50 mm, 65 mm, 35 mm) and (60 mm, 100 mm, 80 mm), respectively. These vertex coordinates are projected onto a PCA basis, and the resulting coefficients $\Lambda = \{\Lambda_1, \Lambda_2, ..., \Lambda_n\}$ are used as shape inputs for training. To improve generalization with limited longitudinal data, we apply data augmentation via random scaling of training shapes by a factor uniformly sampled from the range $[0.98, 1.02]$.

The temporal input $t$ is derived by normalizing the age of subjects. Specifically, we select subjects aged between 65 and 95 years and linearly normalize their age (*i.e.*, $t = (age - 65)/30$ such that $t \in [0, 1]$). The cognitive status $c$ is a continuous value in the range $[0, 1]$, as described in subsection 3.2.

BrainODE is trained for 100 epochs using the AdamW optimizer with a learning rate of 0.0005. For solving the neural ordinary differential equations, we employ the fourth-order Runge-Kutta (RK4) integration method. The training process minimizes the L2 loss function, as derived in Equation 3.

## B  Dataset details

| Dataset | Subjects # | NC | AD | CONV | 1st observed age | # of observ. | Intervals (yrs) |
|---------|-----------|-----|-----|------|------------------|--------------|-----------------|
| LBC1936 | 516 | 516 | 0 | 0 | $73.12 \pm 0.79$ | $3.59 \pm 1.19$ | $3.15 \pm 0.78$ |
| AIBL | 211 | 134 | 56 | 21 | $73.87 \pm 5.25$ | $3.04 \pm 1.15$ | $1.66 \pm 0.57$ |
| ADNI | 829 | 558 | 271 | 0 | $75.35 \pm 5.81$ | $5.05 \pm 3.12$ | $1.07 \pm 0.89$ |
| OASIS2 | 80 | 65 | 11 | 4 | $77.42 \pm 7.11$ | $2.64 \pm 0.78$ | $2.02 \pm 0.70$ |
| OASIS3 | 282 | 253 | 14 | 15 | $71.62 \pm 4.87$ | $2.91 \pm 1.27$ | $2.74 \pm 1.50$ |

Table 5: Composition of longitudinal datasets. NC, AD, CONV denote whose cognitive statuses are normal cognition, Alzheimer's disease, and converted from NC to AD, respectively.

| Validation data | | hippo | | LV | |
|---|---|---|---|---|---|
| Time interval | Dataset | 4shot | 1shot | 4shot | 1shot |
| Regular | LBC | 20 | 20 | 37 | 40 |
| | AIBL | 18 | 19 | 27 | 34 |
| Irregular | AIBL + ADNI + OASIS2 + OASIS3 | 65 | 79 | 63 | 170 |

Table 6: Composition of validation dataset used in subsection 4.1 (Table 1 and Table 2).

### B.1  Dataset Composition

To learn longitudinal brain shape dynamics, we use the LBC1936 [2] and AIBL [3] datasets as regular time interval datasets, whereas AIBL [3], ADNI [6], OASIS2 [10], and OASIS3 [8] are used as irregular time interval datasets. As shown in Table 5, each dataset varies in the number of subjects, age at the first brain image acquisition, number of observations, and intervals between scans.

The LBC1936 data are provided by the Lothian Birth Cohort 1936 Study database[2]. Most participants underwent brain magnetic resonance imaging between 2008 and 2010 at an average age of 72.6 years, with follow-up scans conducted every three years over five waves. The AIBL dataset collects scans every 18 months across five waves. Since AIBL includes subjects with varied ages and provides exact MRI acquisition dates, it is utilized for both regular and irregular time interval analyses. Furthermore, because AIBL, OASIS2, and OASIS3 contain longitudinal data where cognitive status transitions from normal to Alzheimer's disease, their shape data are used for pseudo-cognitive status shape sampling in subsection 3.2.

**Validation data.** The validation dataset composition is Table 6, used for the quantitative results in Table 1 and Table 2. Specifically, we selected subjects whose longitudinal observations are more than four times for the 4-shot prediction.

### B.2  Medical priors in brain shapes.

In this section, we elaborate on the medical priors for longitudinal progression modeling of the brain, as discussed in section 1. Specifically, we analyze the volumes of the hippocampus and LV using all longitudinal datasets (ADNI, OASIS, AIBL, and LBC1936). Additionally, we visualize the rate of volume changes over time intervals to highlight the progression dynamics.

---

[2]https://lothian-birth-cohorts.ed.ac.uk/data-access-collaboration

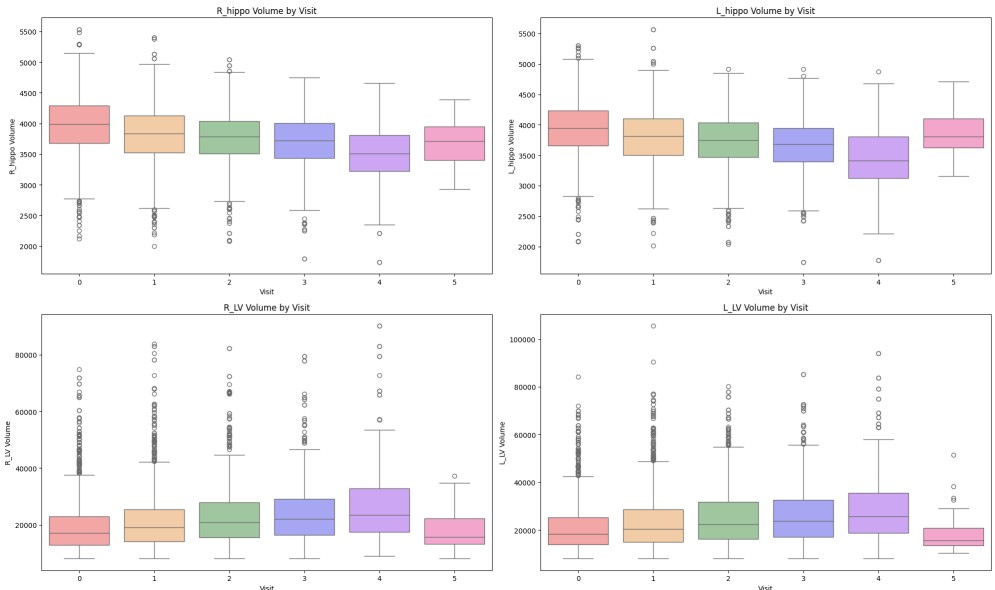

Figure 7: Illustration of brain atrophy progression. (a) Regional volumes across longitudinal observations. (b) Rate of volume changes over time intervals. The first and second rows represent the volumes of the hippocampus and LV, respectively, across all datasets (ADNI, OASIS, AIBL, and LBC1936).

**Brain volume changes.** The human brain undergoes *brain atrophy* (*i.e.*, the loss of brain cells) due to normal aging and neurodegenerative disorders. In particular, Alzheimer's Disease (AD) causes more rapid brain atrophy. As brain atrophy progresses, brain volumes shrink, while the LVs enlarge as the empty space surrounded by brain subcortical structures. Figure 7(a) illustrates brain atrophy and associated volume changes across longitudinal observations for each subject in the brain MRI datasets. Hippocampal volumes show a gradual decline from observation 0 to 4, with the trend less clear at the 5th visit due to limited data availability. Conversely, LV volumes consistently increase over time, reflecting ventricular enlargement caused by the brain atrophy.

To further explore the dynamics of brain atrophy, we calculate the rate of volume changes by dividing the volume difference by the corresponding time intervals (*e.g.*, 1.5 years for AIBL, 3 years for LBC1936, and respective years for irregular time intervals). This rate, expressed in $mm^3$/year, quantifies the speed of atrophy or enlargement. Figure 7(b) visualizes these rates across age groups, defined based on the dataset distributions. The hippocampus shows a higher rate of volume reduction in older age groups compared to the 47–57 group, whereas LV enlargement accelerates with age. Notably, since older brains are smaller in volume, a similar absolute rate of volume loss implies relatively more severe atrophy in later life stages (*e.g.*, 76–86 vs. 86–96 age groups). These trends align with the medical priors, confirming that older age exacerbates the progression of atrophy.

To assess the impact of cognitive status, Figure 8 compares the rate of volume changes between normal cognition (NC) and AD in the same age groups. We visualize the change rate of the left and right brain regions separately. AD (red) exhibits a significantly faster and more irregular rate of volume change than NC (blue). For example, in the 67–76 age group, the LV volume reduction rate is approximately -700 $mm^3$/year for NC, compared to -1600 $mm^3$/year with greater variability in AD. This heterogeneous pathological progression of AD presents challenges for modeling as it requires capturing complex and non-linear patterns that diverge from the more stable trends observed in NC.

## C    Additional analysis and visualization

In this section, we comprehensively evaluate longitudinal brain shape prediction on multiple perspectives. We first present additional experimental results from subsection 4.1, including the qualitative results of the 1-shot prediction in subsection C.1, qualitative performance across benchmarks in subsection C.2, the effectiveness of the pseudo-cognitive status embedding in subsection C.3, and

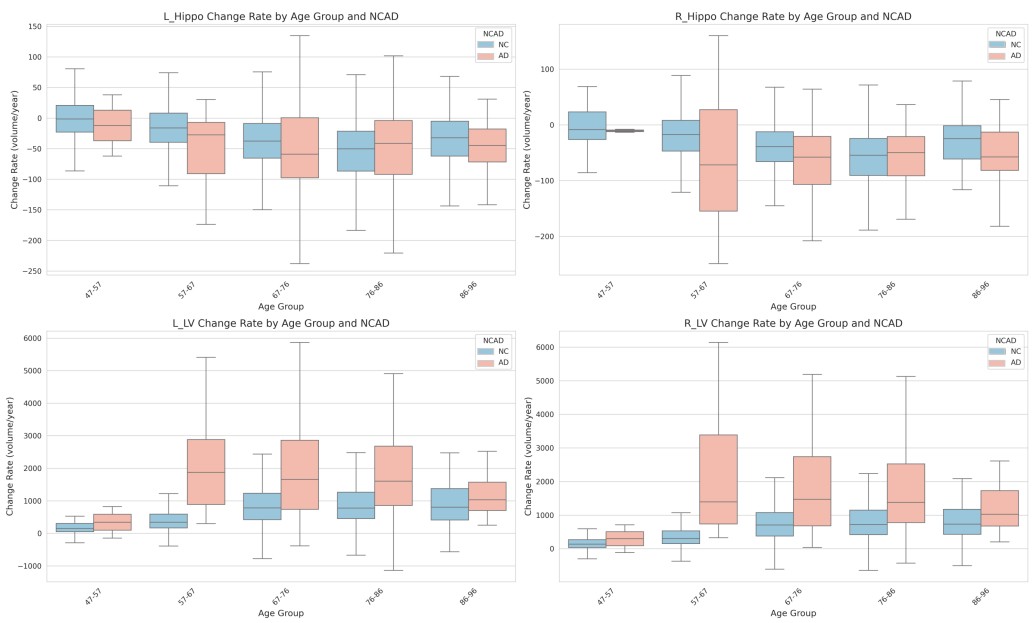

Figure 8: Rate of volume changes over time intervals for normal cognition (NC) and Alzheimer's Disease (AD). The age groups are split based on the data distribution from ADNI, OASIS, AIBL, and LBC1936. NC and AD groups are visualized in blue and red, respectively

BrainODE's condition fidelity in modeling Alzheimer's Disease (AD) in subsection C.4. Furthermore, we examine the subject inconsistency inherent in image generative approaches for brain region progression modeling compared to deformation-based methods in subsection C.5, compare the prediction performance with registration-based method in subsection C.6. Finally, we evaluate the long-range shape prediction capabilities of baseline methods and BrainODE in subsection C.7.

## C.1 Additional analysis of experiments

We present qualitative results for brain shape modeling using 1-shot prediction in Figure 9. Unlike the 4-shot setting in Figure 2, which leverages multiple time points to guide the deformation trajectory, the 1-shot setting lacks temporal deformation information. As a result, conventional extrapolation methods are excluded.

**Qualitative evaluation of 1-shot prediction.** To implement prediction on arbitrary time points using ShapeFlow [7], which does not support irregular time intervals, we train the model on the AIBL [3] dataset using fixed 1.5-year intervals. The starting time point is set to $t = 0$, and 1.5 years is mapped to $t = 1$. Consequently, future time targets are normalized relative to this interval (*e.g.*, a 2-year prediction corresponds to $t = 2/1.5$).

Other baseline methods that can handle irregular time intervals are also included for comparison. This setup allows us to evaluate the ability of each method to predict future shapes under varying temporal gaps. Notably, addressing irregular time intervals is critical in longitudinal medical imaging, as patient visit schedules often vary. Robust modeling across inconsistent time gaps is essential for accurately capturing disease progression.

Figure 9 visualizes qualitative results of 1-shot prediction for hippocampus and LV across varying ages and cognitive statuses, including normal cognition (NC) and Alzheimer's disease (AD). Overall, BrainODE outperforms baseline methods in modeling both brain regions. Specifically, for the hippocampus, both BrainODE and ShapeFlow achieve small Euclidean distances from the ground truth. However, BrainODE uniquely excels in LV prediction, demonstrating the lowest errors.

Regarding the baseline results, ShapeFlow accurately predicts hippocampus shapes but struggles to model the LVs, showing significant errors (> 4 mm) in several regions. LatentODE [15] produces incorrect LV shapes by altering their topology and fails to capture subject-specific features, generating

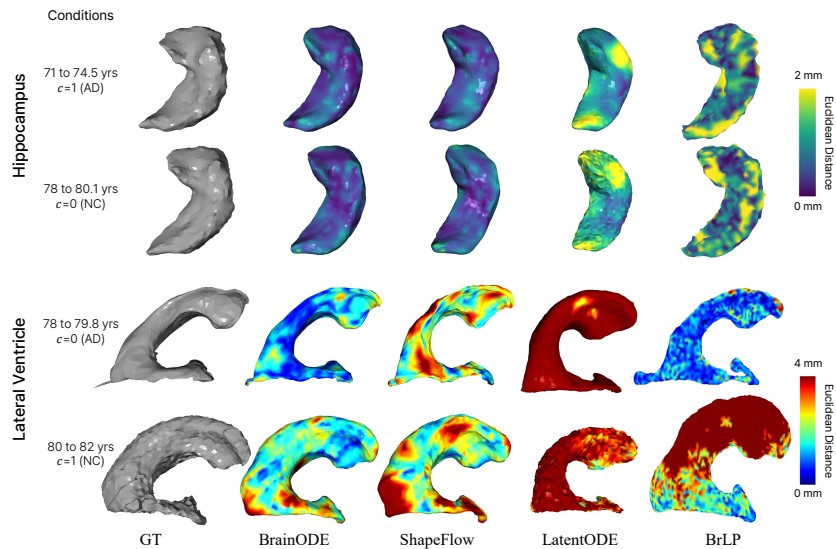

Figure 9: Qualitative results of 1-shot prediction for hippocampus and LV shapes. The color map indicates Euclidean distance errors (up to 2 mm for hippocampus, 4 mm for LV) between ground truth (GT) and predicted shapes.

nearly identical shapes regardless of input conditions. This suggests that LatentODE struggles to model the prior distribution of brain progression dynamics using autoencoders, leading to statistically plausible yet trivial solutions (*e.g.*, converging to average shapes) that disregard the given inputs. Similarly, BrLP distorts the anatomical features of the input shapes. For instance, over-enlargements are observed on the occipital horn, frontal horn, and body of the LV in NC subjects, and topological changes appear in occipital horn of the AD subject. In contrast, despite the challenges of modeling brain progression with irregular time intervals, BrainODE consistently demonstrates superior performance, validating its robustness and accuracy.

## C.2 Qualitative evaluation across benchmarks

| Dataset | 4-shot | 1-shot |
|---------|--------|--------|
| AIBL | $0.52 \pm 0.07$ | $0.46 \pm 0.05$ |
| ADNI | $0.59 \pm 0.14$ | $0.54 \pm 0.15$ |
| OASIS | $0.52 \pm 0.07$ | $0.48 \pm 0.08$ |

Table 7: Quatitative evaluation of hippocampal shape prediction performance in Euclidean distance (mm) on datasets with irregular time intervals.

The detailed qualitative results in Table 2 by datasets are illustrated in Table 7. From these results, although the discrepancies are not substantial, it underscores the need for further validation across benchmarks. To this end, we conducted the following cross-benchmark experiments.

For Exp1, BrainODE is trained on AIBL (with regular time intervals) and validated on ADNI and OASIS (with irregular time intervals). For Exp2, BrainODE is trained on AIBL, ADNI and OASIS (with irregular time intervals) and validated on LBC1936 (with regular time intervals).

These experiments allow us to evaluate the robustness of BrainODE under both temporal and benchmark distribution shifts. As shown in Table 8, while performance under these shifts is slightly degraded compared to intra-dataset benchmarks, BrainODE consistently outperforms other baseline models, demonstrating robustness to unseen individual shape morphologies and progression patterns. In terms of generalizability, each cohort (dataset) follows different temporal sampling intervals, ranging from 0.5 years (e.g., ADNI) to over 3 years (e.g., LBC1936). Due to this temporal discrepancy, BrainODE exhibits performance decreases in both shift scenarios. These findings underscore the importance of training on large-scale longitudinal datasets with varied time intervals to enhance

generalization across cohorts. Leveraging broader and more diverse training data is essential for improving robustness and capturing realistic progression dynamics.

Table 8: Cross benchmark validation results for hippocampus.

|  | Train | Validation | Euclidean dist. (mm) |
|---|---|---|---|
| Exp1 | AIBL | ADNI + OASIS | 0.518 |
| Exp2 | AIBL + ADNI + OASIS | LBC1936 | 0.522 |

## C.3 Sampling longitudinal shapes with intermediate timepoints

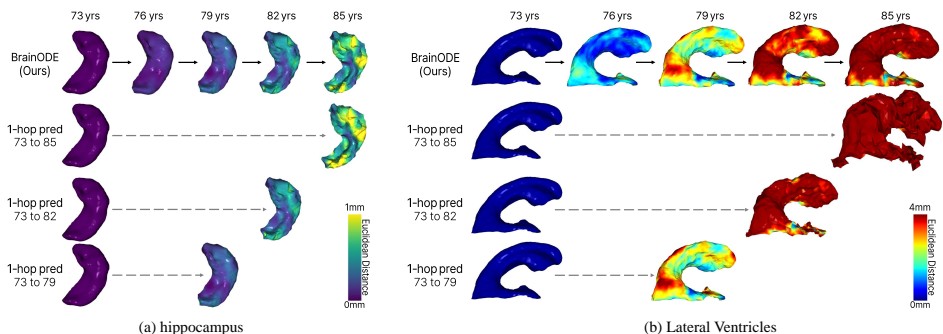

Figure 10: Qualitative results of sampling longitudinal shapes with intermediate time points for (a) hippocampus and (b) LVs.

In this section, we explore the effects of incorporating pseudo-cognitive status embeddings and generating shapes on intermediate time points to enable progressive longitudinal modeling. Specifically, we investigate how this approach impacts the accuracy and stability of shape predictions. To this end, we compare the qualitative results of shapes generated every 3 years over a 12-year period with those produced in a single step (*i.e.*, 1-hop prediction).

Figure 10 illustrates the results, showing the deformation of hippocampus and LV shapes from age 73 to 85, along with the Euclidean distance maps from the initial shape at age 73. Predicting brain shapes over long intervals in a single step is challenging for BrainODE, as such large temporal gaps are not seen during training, where the average interval between the first and last acquisitions is $4.08 \pm 2.84$ years. To address this, BrainODE adopts an iterative prediction strategy, forecasting future shapes through intermediate time steps, as shown in the first row of the figure.

Direct prediction over a 12-year interval results in unrealistic deformations and surface artifacts; especially, the LV surface is almost entirely destroyed. In contrast, when intermediate predictions are made every 3 years, the final 12-year prediction maintains anatomical consistency and surface smoothness. This experiment highlights the effectiveness of progressive pseudo-shape sampling in modeling longitudinal brain progression.

## C.4 Tendency of shape changes by age and cognitive state

To further demonstrate the *condition-injectivity* of BrainODE, we visualize shape predictions under varying cognitive statuses $c = \{0, 0.5, 1.0\}$ in Figure 11. BrainODE predicts larger deformations in both the hippocampus and LVs as cognitive status declines, consistent with clinical progression patterns observed in longitudinal data (subsection B.2). For instance, the shape predicted with intermediate status $c = 0.5$ shows deformations that lie between those of normal cognition and Alzheimer's disease, as also reflected in their estimated volumes.

Beyond qualitative visualization, we analyze volumetric trends across varying cognitive statuses for 100 baseline hippocampal

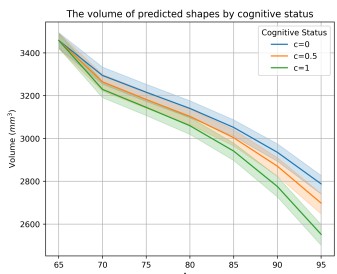

Figure 12: Brain Atrophy tendency by cognitive status $c$ modeled by BrainODE.

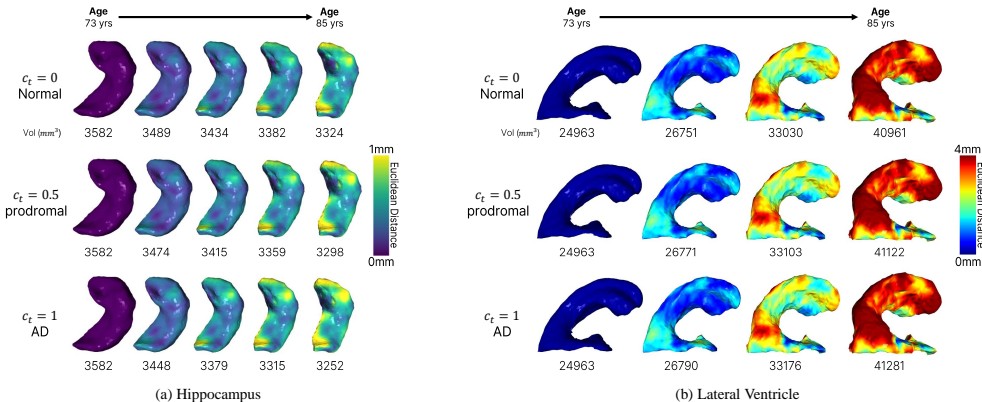

(a) Hippocampus          (b) Lateral Ventricle

Figure 11: Longitudinal shape predictions with varying cognitive status $c = \{0, 0.5, 1\}$ every 3 years for hippocampus and LV. The number below each shape denotes the volume of the predicted shape.

shapes in Figure 12 at the age of 65. Even though explicit medical priors is not injected during training, BrainODE captures clinically plausible volume trajectories by leveraging the spatiotemporal supervision from longitudinal datasets. This supports the ability of BrainODE to encode implicit medical priors through learning dynamics across both age $t$ and cognitive status $c$.

## C.5 Subject inconsistency of generative approaches

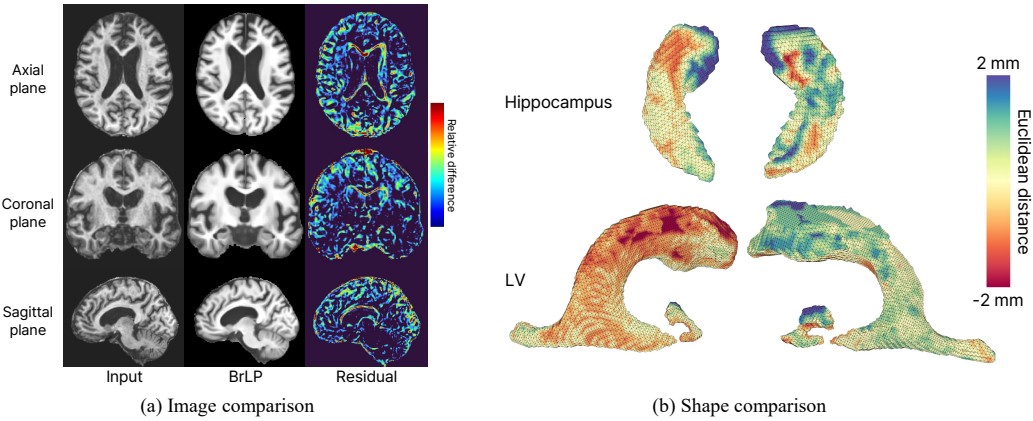

(a) Image comparison          (b) Shape comparison

Figure 13: Subject inconsistency of the BrLP [12] generative model. (a) Image comparison shows the input brain MRI scan (axial, coronal, sagittal planes) and the residual difference after BrLP generation with $\Delta t = 0$. (b) Shape comparison highlights Euclidean distance errors (up to $\pm 2$ mm) between the input and generated hippocampus and LV shapes.

In this study, we examine two primary approaches for modeling longitudinal progression in neurodegenerative diseases: generative models and deformation-based methods, which exhibit distinct characteristics. Generative approaches, such as BrLP [12], model medical priors using the distribution of given data and its associated conditions (*e.g.*, age and cognitive status). During inference, these models generate statistically plausible samples from the learned distribution. However, as discussed in subsection 4.1, we hypothesize that this approach is unsuitable for longitudinal progression modeling of medical data, as it is challenging to ensure spatial consistency of each subject (*i.e.*, maintaining the sampled data's identity).To test this, we conducted a preservation experiment using the BrLP generative model by inputting a brain MRI scan with no time difference ($\Delta t = 0$) and measuring the differences between the input and generated outputs in both image and shape spaces, as depicted in Figure 13.

In the image-level comparison (Figure 13(a)), although the output is expected to be identical to the input, noticeable differences are observed in high-frequency details and anatomical structures such as the LV, as highlighted by their residuals in the color map. We further visualize shapes of the hippocampus and LVs, which are of two key brain regions to determine the cognitive status, in the shape-level comparison (Figure Figure 13(b)). The Euclidean distance is calculated with the input subject's shapes, indicating significant deviations in the hippocampus and LV shapes. In addition, the LV has a seamless structure in the brain, however, BrLP generates a disconnected LV due to limited voxel grid resolution. These inconsistencies could compromise longitudinal diagnosis, potentially perturbing the AD decision boundary of deterministic classification models sensitive to subtle changes, as reported in [9]. This highlights a fundamental limitation of generative models in maintaining subject-specific fidelity. In contrast, deformation-based approaches such as BrainODE offer improved consistency by preserving anatomical identity when $\Delta t = 0$, making them more suitable for reliable longitudinal modeling.

## C.6 Comparison with registration-based method

We additionally experimented with NODER [3], which aims to predict missing timepoints in brain image sequences. NODER requires at least two prior timepoints as input for sequence regression, which limits its practical applicability for predicting future shapes from a single observation. More-over, it requires separate training for each individual brain image sequence, making it unsuitable for learning generalizable longitudinal dynamics.

We conducted this experiment using the LBC1936 dataset under a 4-shot validation setting. Although NODER utilizes deformation information from earlier timepoints, its voxel-based regression approach and lack of explicit conditioning on age and disease lead to inferior predictive accuracy. This highlights the effectiveness of BrainODE's design, which captures shape dynamics among populations in mesh representation and explicitly conditions on age and cognitive status, enabling more accurate modeling of longitudinal shape changes.

Table 9: Comparison the hippocampal shape prediction performance in Euclidean distance (mm) between registration-based NODER [3] and ours on the LBC1936 dataset.

| Method | hippo | LV |
|---|---|---|
| NODER [3] | 0.898 | 4.152 |
| Ours | **0.488** | **1.630** |

## C.7 Longitudinal shape prediction capacity

From a practical perspective, an important question arises in longitudinal modeling: how far into the future can models reliably predict brain shape progression while maintaining anatomical consistency? To investigate this, we evaluated the longitudinal prediction capabilities of five models—ShapeFlow, Extrapolation, BrLP, LatentODE, and BrainODE—using the LBC1936 dataset, as shown in Figure 14. The figure presents the predicted hippocampus shapes of a single subject spanning ages 65 to 95 (within the training distribution) and extending to 107 years (beyond the training distribution), demonstrating each model's ability to predict long-term brain shape progression.

Specifically, ShapeFlow fails to predict beyond 89 years, as the brain shapes "explode", making visualization infeasible. Extrapolation, while able to generate shape predictions, shows progressive collapse when forecasting further beyond the two input time points (*e.g.*, 65 and 71 years), with no-ticeable distortions at 95 years. BrLP appears plausible at a glance, but as discussed in subsection C.5, exhibits subject inconsistency; the predicted shape at 71 years already shows changes in overall geometry and new appearances, not in the shape at 65 years. LatentODE, consistent with the main experimental results, struggles with prior modeling within its VAE framework, repeatedly predicting similar shapes (*e.g.*, minimal variation in LV shape across all ages) due to training difficulties. In contrast, BrainODE demonstrates robust performance within the in-distribution age, progressively transforming shapes while preserving anatomical geometry features, such as the curvature of the hippocampus and the expansion patterns of the LV, even up to 95 years. This highlights superior capacity of BrainODE for long-term longitudinal prediction compared to other models.

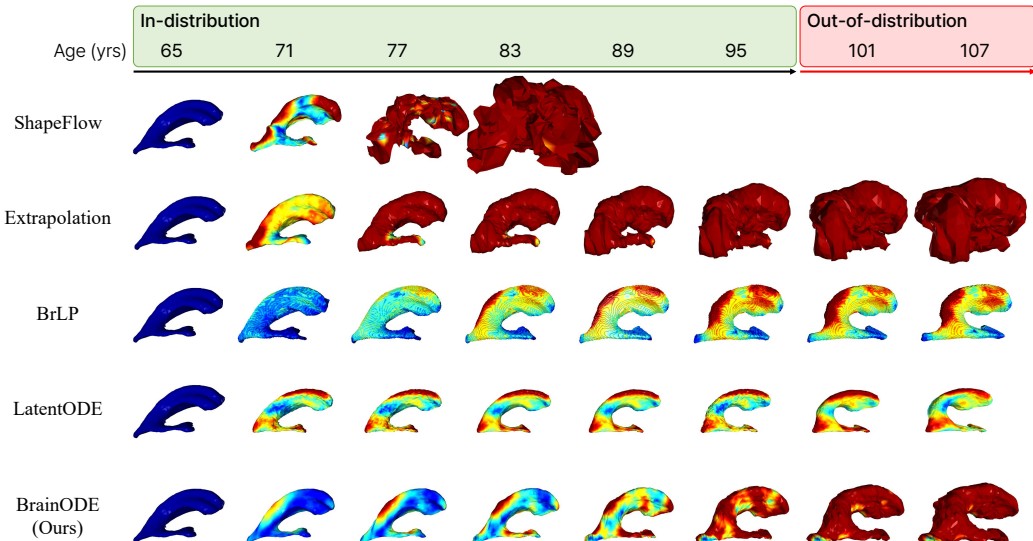

Figure 14: Comparison of longitudinal shape prediction over time for the hippocampus and LV. Each column represents a different model, showing shapes predicted from ages 65 to 107 at uniform intervals. Colors indicate the Euclidean distance from the first shape at age 65.

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
