# OpenReview forum: "BrainODE: Neural Shape Dynamics for Age- and Disease-aware Brain Trajectories"
_NeurIPS.cc/2025/Conference — NeurIPS 2025 poster_

### Official Review · Reviewer_6kif · 2025-06-10

**Clarity:** 3
**Significance:** 3
**Originality:** 3
**Rating:** 5
**Confidence:** 4

**Summary:**

The authors attempt to model the age-related and disease-related deformation of brain anatomy over time. The proposed method, BrainODE, is based on neural ordinary differential equations (neural ODEs). BrainODE modulates shape dynamics with subject-specific age and cognitive status, and is able to forecast from a single observation autoregressively over an embedding space that jointly encodes age and estimated cognitive status. The authors performed quantitative experiments on several brain imaging datasets with subjects following normal aging and cognitive diseases, covering use cases with regular and irregular time intervals. BrainODE shows competitive results compared to simple extrapolation and autoregressive baselines, as well as more relevant methods such as LatentODE and BrLP.

**Questions:**

I have 3 actionable suggestions that likely does not require additional experiments.

1. See Weakness 1a
2. See Weakness 1b
3. See Weakness 1c

**Ethical Concerns:**

["NO or VERY MINOR ethics concerns only"]

**Final Justification:**

In the initial review, I found this work quite interesting but undersold in the current form of presentation, and gave a handful of suggestions regarding presentation and results to report. The authors did a decent amount of work to nicely address all of them. I have also read the reviews of other reviewers, and I personally found them to be sufficiently addressed by the authors.

**Limitations:**

I recommend the authors to include a section to discuss the Limitations. I do not foresee any potential negative societal impact.

**Paper Formatting Concerns:**

1. I believe by convention the caption of a table should appear above instead of below the table.

**Quality:**

3

**Strengths And Weaknesses:**

Strengths:
1. The authors are tackling an interesting and potentially high-impact problem. The solution they proposed – modeling disease progression using a variant of neural ODE – is very intuitive.

2. The use of shapes and meshes instead of modeling in the 3D voxel space is fairly well explained.

3. I particularly like the visualizations (Figure 2 and 3) where the error in Euclidean distance is highlighted using a visually distinguishing color map.

4. The number and diversity of the datasets chosen for experiments are decent. The suite of baseline methods for comparison is fairly reasonable. The quantitative performances look okay.

Weaknesses:
1. The presentation of the results can be made stronger, likely without additional experiments.

    1a. Table 1 and 2 could include additional metrics beyond the error in Euclidean distance: for example, (1) image/mesh similarity metrics such as PSNR, (2) area-of-interest overlap metrics such as dice coefficient, (3) surface smoothness metrics – but note surface smoothness should be compared in reference of the ground truth since it wouldn’t be the smoother the better. I am fairly sure BrainODE can beat the competing methods judging from the other results, but these would provide more comprehensive views of the superiority.

    1b. Table 2 could (and should) be subset into the datasets (ADNI, OASIS and AIBL).

    1c. While Figure 3 looks cool, the authors can potentially enhance it. In my opinion, the authors can additionally show how BrainODE generates smoother trajectories compared to other methods. I would suggest keeping these 4 subfigures as-is as panel A, and additionally include a panel B below, showing line plots quantifying these trajectories, and also include other methods’ trajectories for comparison. Line plots can be grouped by method or grouped by the 4 conditions.

2. There are several prior works in this area that have not been discussed. For example, the most closely related might be ImageFlowNet [1], which uses an adapted neural ODE/SDE to model disease progression in a joint embedding space. Since they do not seem to have a native implementation in 3D volume due to computational cost, I wouldn’t urge the authors to include it for comparison given the amount of additional work. Similarly, a few other techniques have been used to model the progression of diseases in images or volumes, including flow matching [2] and diffusion models [3] [4]. Again, I am not asking for additional experiments, but it would be beneficial to discuss these works and highlight how BrainODE offers a distinct (and more favorable) solution in this context.

[1] ImageFlowNet: Forecasting Multiscale Image-Level Trajectories of Disease Progression with Irregularly-Sampled Longitudinal Medical Images, ICASSP 2025 Oral.

[2] Modeling Complex System Dynamics with Flow Matching Across Time and Conditions, ICLR 2025 Spotlight.

[3] Denoising diffusion probabilistic models for 3D medical image generation, Scientific Reports 2023.

[4] SADM: Sequence-Aware Diffusion Model for Longitudinal Medical Image Generation, IPMI 2023.

---

> ### Author Rebuttal · Authors · 2025-07-28
>
> We sincerely appreciate your suggestions (W1) and insightful comments on related works (W2), which have guided us in enhancing the clarity of our paper.
> ***
> >### (W1a)
>
> We agree with your suggestion regarding the inclusion of additional voxel-based evaluation metrics or mesh quality assessments.
>
> In this work, we initially omitted voxel-based metrics because our method operates directly in the 3D mesh domain. In addition, while voxelization of the hippocampus is straightforward, the lateral ventricle (LV) presents unique challenges: despite its anatomically connected structure, ventricular masks in MRI data are often fragmented or disconnected due to limited resolution. Although we address this by leveraging mesh reconstruction, it makes voxel-based comparisons with mesh occupancy less reliable for LV structures.
>
> Nevertheless, as you pointed out, voxel-based metrics can enhance interpretability and comparability with voxel-based approaches. To provide a complementary perspective, we computed **Dice scores, widely used in medical imaging, for the hippocampal shapes.** These results, presented below and to be included in the appendix. The relative ranking of alignment quality remains consistent with Tab 1&2 as you anticipated.
> | Method    | LBC1936 4-shot | LBC1936 1-shot | AIBL 4-shot | AIBL 1-shot |
> |-----------|----------------|----------------|-------------|-------------|
> | ShapeFlow | 0.90 ± 0.02    | 0.92 ± 0.04    | 0.89 ± 0.04 | 0.92 ± 0.02 |
> | BrLP      | 0.84 ± 0.03    | 0.84 ± 0.03    | 0.87 ± 0.03 | 0.86 ± 0.05 |
> | Ours      | **0.93 ± 0.02**    | **0.94 ± 0.01**    | **0.93 ± 0.01** | **0.94 ± 0.01** |
> ***
> >### (W1b)
>
> We agree with your suggestion to provide subset accuracy for Table 2 and will incorporate the results into the final version of the paper.
> For hippocampal shape analysis, the number of subjects in the 4-shot is 18, 20, and 27, and in the 1-shot is 19, 20, and 40, for AIBL, ADNI, and OASIS respectively. The subset results on our method are presented below, and we will incorporate this to the final version.
> | Dataset | 4-shot       | 1-shot       |
> |---------|--------------|--------------|
> | AIBL    | 0.52 ± 0.07  | 0.46 ± 0.05  |
> | ADNI    | 0.59 ± 0.14  | 0.54 ± 0.15  |
> | OASIS   | 0.52 ± 0.07  | 0.48 ± 0.08  |
>
> From these results, although the discrepancies are not substantial, it underscores the need for further validation across benchmarks. To this end, we conducted the following additional experiments:
>  -  Exp1: Training on AIBL (with regular time intervals) and validating on ADNI and OASIS (with irregular time intervals).
>  - Exp2: Training on AIBL + ADNI + OASIS (with irregular time intervals) and validating on LBC1936 (with regular time intervals).
>
> These settings allow us to evaluate the robustness of BrainODE under both temporal and benchmark distribution shifts. While performance under these shifts is slightly degraded compared to intra-dataset benchmarks, BrainODE consistently outperforms other baseline models, demonstrating robustness to unseen individual shape morphologies and progression patterns.
>
> In terms of generalizability, each cohort (dataset) follows different temporal sampling intervals, ranging from 0.5 years (e.g., ADNI) to over 3 years (e.g., LBC). Due to this temporal discrepancy, BrainODE exhibits marginal performance decreases in both shift scenarios.
>
> These findings underscore the importance of training on large-scale longitudinal datasets with varied time intervals to enhance generalization across cohorts. Leveraging broader and more diverse training data is essential for improving robustness and capturing realistic progression dynamics. We plan to include this validation in the final version of our work.
>
> | Experiment | Training                | Validation     | Hippo    | LV       |
> |:------------:|:-------------------------:|:-------------------:|:-------------------:|:-------------------:|
> | Exp1       | AIBL                    | ADNI + OASIS   | 0.518 (↑0.016) | 2.265 (↑0.554) |
> | Exp2       | AIBL + ADNI + OASIS     | LBC1936        | 0.522 (↑0.172) | 1.988 (↑0.203) |
>
> (Values in parentheses are the surface distance metric (mm) difference from the intra-benchmark result)
> ***
> >### (W1c)
>
> We are truly grateful for your suggestion regarding improved visualization of our results. In response, we added smoother trajectory visualizations compared to the baselines in Fig 14 of Appendix C.5. Additionally, your suggestion to use line plots is an insightful idea for highlighting the smooth longitudinal patterns.
>
> Following your suggestion, we will include a smoothed trajectory visualization in Figure 3 to better illustrate the advantages of our method.
> ***
> >### (W2)
>
>
> We appreciate your suggestion to include related works and agree that discussing these models would help better define the scope of our paper. Below, we provide a brief overview of how BrainODE distinguishes itself, particularly in terms of representational effectiveness, computational efficiency, and subject-level modeling.
>
> - ImageFlowNet [1] models subject-specific spatiotemporal dynamics in 2D medical images using neural ODEs/SDEs with multi-scale representations based on a U-Net architecture. Since it performs progression modeling in the pixel space, it presents several limitations in computational cost and precision. The use of multi-scale representations in a joint space of neural ODEs/SDEs and pixel-level reconstruction results in substantial memory consumption, making it infeasible in 3D voxel space (e.g., over 80 GB for volumes of size 128^3). Furthermore, its ability to capture subtle changes, crucial for modeling disease progression, is inherently limited by the resolution of the input image due to the discrete nature of the pixel grid.
> In contrast, BrainODE employs mesh representations and PCA-based 3D shape modeling, which significantly reduces computational requirements (~375 MB for training) while achieving high-precision shape modeling with a ~0.4 mm average error for hippocampus.
>
> - Multi-Marginal Flow Matching (MMFM) [2] shares several core similarities with our BrainODE. Both methods employ a single shared model to learn across diverse conditions, support continuous dynamics modeling, handle irregular and multi-time-point data, and incorporate techniques for imputing mission observation.
> However, MMFM is designed to model distributional translations, making it well-suited for capturing general patterns and group-level dynamics. Therefore, it does not predict explicit individual-level trajectories and does not incorporate explicit structural parameters. In contrast, BrainODE embeds brain morphology directly into a parameter space and models continuous shape trajectories at both individual and disease group levels. This enables precise tracking and predicting of morphological changes for each subject.
>
> - As we discussed in the main paper, image generative model-based approaches ([3], [4] and BrLP) have several limitations: substantial computational costs, inconsistencies in subject identity due to Gaussian sampling (i.e., subject-specificity is lost due to stochasticity in generation), coarse resolution that fails to capture subtle yet clinically important changes. Furthermore, none of these methods is capable of modeling the clinical continuum of the AD trajectory as they rely on discrete condition values for the disease labeling.
> ***
> >### (L1 & Paper formatting)
>
> - We agree with the reviewers' suggestions to move the discussion of limitations from the appendix to the main paper, thus, we plan to include it in the discussion section. This will help appropriately acknowledge the boundaries of our work and clarify where this pioneering effort in longitudinal brain shape prediction is positioned within the medical imaging field.
> - Thank you for pointing out the issue with the table caption. We will adjust its placement accordingly in the final version.

---

> > ### Comment · Reviewer_6kif · 2025-07-31
> > **Response to rebuttal**
> >
> > Dear authors,
> >
> > Thank you for the comprehensive point-by-point response. With these efforts, my concerns have been addressed and do not have further questions at the moment.

---

> > > ### Author Response · Authors · 2025-08-09
> > >
> > > We sincerely appreciate your insightful feedback. We will carefully refine and clarify our work in light of your comments. Thank you for your guidance throughout the review process.

---

### Official Review · Reviewer_4KRF · 2025-07-02

**Clarity:** 2
**Significance:** 2
**Originality:** 3
**Rating:** 4
**Confidence:** 5

**Summary:**

This paper presents BrainODE, a neural ODE framework for modeling brain shape trajectories conditioned on age and cognitive status. Operating in a PCA space of 3D meshes (hippocampus and ventricles), the model predicts anatomically plausible deformations from sparse longitudinal data. A key contribution is the pseudo-cognitive status embedding, enabling interpolation along the cognitive decline spectrum. Evaluated on four neuroimaging datasets, BrainODE outperforms RNN, latent ODE, and flow-based baselines in both one-shot and multi-shot settings.

**Questions:**

1. The authors claim strong generalization, but the method is demonstrated only on the hippocampus and lateral ventricles. Could BrainODE be readily extended to other brain regions or full cortical surfaces? Are there computational or modeling limitations that prevent this? Additionally, how can BrainODE be applied to external studies? For example, if the model is trained on studies A and B, how would it perform on study C? Such an analysis is critical to substantiate the claim of "strong generalization".

2. Can the authors elaborate on how reliable the pseudo-cognitive status embedding is when applied to unseen subjects? How sensitive is BrainODE’s prediction to errors in the cognition estimator? A sensitivity analysis would help clarify the robustness of this mechanism and strengthen the reader’s confidence in the method’s applicability.

3. Why are MCI subjects excluded from the modeling? Given that MCI is a clinically critical intermediate stage, how do the authors justify its exclusion beyond citing ambiguous labeling? The authors emphasize the practicality of their method, but omitting an entire diagnostic class—particularly one central to early-stage Alzheimer's detection—substantially undermines this claim.

4. Is there a reason why the authors do not compare with ConDOR although they mention it in the prior work?

5. Why do the authors pick the 4-shot prediction and do not experiment with 2-shot, 3-shot etc ? Gradually increasing subject information rather than moving directly to the 4-shot would strengthen the paper.

**Ethical Concerns:**

["NO or VERY MINOR ethics concerns only"]

**Final Justification:**

The authors addressed most of my concerns. However, My main concern regarding the handling of the MCI class as well as the sensitivity of the cognition estimator remains. The authors mentioned that will include the sensitivity analysis in the manuscript but I am not able to evaluate it for now as I do not have access to the updated manuscript.  However, my position for this paper tends to be positive, and thus I decide to increase my score to 4.

**Limitations:**

The authors mention key limitations, but only in the appendix rather than in the main paper. They acknowledge the exclusion of MCI subjects due to ambiguous labeling and discuss future directions for integrating this critical diagnostic group. However, given MCI’s clinical importance and its role in early intervention studies, this limitation should be more prominently discussed. Bringing these limitations into the main text and expanding on their practical implications in case of misdiagnosis would strengthen the paper’s clinical impact. Additionaly the authors. In the limitations section, the authors mention that they average predicted shapes across timepoints for subjects with multiple observations and highlight the need for more advanced aggregation strategies. I encourage the authors to acknowledge recent work on modeling longitudinal trajectories with repeated measures, such as the Adaptive Shrinkage Estimation for Personalized Deep Kernel Regression (ICLR 2025), which addresses the problem of longitudinal prediction of neurodegeneration with repeated measures.

**Paper Formatting Concerns:**

No concerns regarding the formatting.

**Quality:**

2

**Strengths And Weaknesses:**

**Strengths**
The paper addresses an important clinical problem—modeling longitudinal brain shape changes related to neurodegeneration—which has clear relevance for Alzheimer’s research. The authors conduct quantitative and qualitative evaluations on multiple datasets, showing that their model performs favorably compared to selected baselines.

**Weakness**
1. The paper would benefit from clearer organization and a more unified presentation of the experimental setting. In particular, the distinction between regular and irregular time intervals seems unnecessary; irregular sampling is a general case that naturally subsumes the regular setting. Consolidating the analysis under this more realistic and flexible framework could simplify the narrative and strengthen the practical relevance of the work. Additionally, the manuscript lacks important details regarding the construction of training and test splits across datasets, which limits reproducibility and interpretability of the results. The absence of an appendix in the main paper reduces accessibility to key implementation specifics. Finally, the quantitative evaluation could be significantly improved by reporting results over multiple runs, along with confidence intervals or statistical significance testing, to support the robustness of the performance claims.
2.  The authors claim strong generalization, but the method is evaluated only on the hippocampus and lateral ventricles. It remains unclear whether BrainODE can scale to more complex anatomical regions or cortical surfaces, and whether there are architectural or computational bottlenecks that limit such extensions. Moreover, there is no assessment of the model’s ability to generalize across studies. Evaluating performance on an unseen dataset (e.g., training on ADNI/AIBL and testing on OASIS) would be critical to support the generalization claims.
3.  The authors exclude MCI subjects from the modeling due to ambiguous labeling, but MCI represents a clinically important transitional stage with significant relevance to early detection and intervention. Omitting this class substantially reduces the practicality and applicability of the method in real-world clinical settings.
4.  The evaluation considers 1-shot and 4-shot prediction settings, but does not explore intermediate scenarios (e.g., 2-shot or 3-shot). Gradually increasing the available subject information would provide a more complete picture of model performance and help assess how BrainODE scales with longitudinal depth.

---

> ### Author Rebuttal · Authors · 2025-07-30
>
> >### (W1a) Regular vs. irregular time intervals
>
> We distinguished regular and irregular time intervals **to evaluate BrainODE under varying settings**. It allows broad comparisons with baselines: RNN, LSTM, and ShapeFlow are suitable for regular time series and RNN-Decay, LatentODE, and BrLP can handle irregular intervals. BrainODE is designed to handle both, and we report performance respectively.
>
> >### (W1b)Lack of detail in dataset splits and reproducibility & statistical robustness in evaluation**
>
> For reproducibility, we have **shared the implementation repository**(L415). While we are unable to share the dataset itself due to data usage policies, we provide all relevant code and a toy example to facilitate reproducibility. In addition, we described dataset split details in Tab5&6.
> For statistical robustness, we appreciate that sharing your concerns for statistical rigor in our evaluation protocol. Our protocal for Tab1&2 is:
>   1) For 1-shot prediction, we performed three validation rounds with different groups. As a result, their ranking was not changed across validations.
>   2) For 4-shot prediction, since subjects with ≥5 time points are rare, we evaluated on all such subjects of all benchmarks.
>   3) For the pseudo-cognitive status shape sampling experiments (Sec 3.2 & Tab. 3), we conducted leave-four-out cross validation three times using 40 CONV-labeled subjects.
>
> To clarify, we originally omitted statistical values(e.g., std, significance) since heterogeneous time intervals across validation folds could confound accuracy interpretations. Instead, we focused on **reporting consistent trends in model ranking, which were preserved across multiple resampling scenarios.** But for clarity, we will offer standard deviation for better interpretation following your suggestion. In addition, we will include above validation protocols in the final version.
> ***
> >### (W2&Q1)
> ### [Generalization to brain structures]
> The reason why we focus on two subcortical structures is not only because they are highly relevant to AD but also due to their distinct morphological characteristics. The hippocampus is a critical region for AD diagnosis, occupying about 1.7%  of the total brain volume and reflecting conspicuous localized atrophic changes. In contrast, LV serves as a global indicator of brain atrophy by its enlargement with the large-scale morphological variations.
>
> As the reviewer points out, **BrainODE can be readily extended to other subcortical region** by applying the same pipeline in Appendix A.1. However, extending to cortex presents additional challenges due to their complex folded geometry and establishing reliable point correspondences across inter- and intra- subjects. To model longitudinal cortex shape prediction, tailored methods such as CortexODE(2022) could be integrated with BrainODE. We will include this discussion in the final version.
>
> ### [Generalization across datasets]
> We agree that validating generalization performance across datasets is an important direction, so we conducted the following additional experiments for 1-shot prediction:
>  -  Exp1: Training on AIBL (with regular time intervals) and validating on ADNI and OASIS (with irregular time intervals).
>  - Exp2: Training on AIBL + ADNI + OASIS (with irregular time intervals) and validating on LBC1936 (with regular time intervals).
>
> These settings allow us to evaluate the robustness of BrainODE under both temporal and benchmark shifts. While performance under these shifts is slightly degraded compared to intra-dataset benchmarks, BrainODE consistently outperforms other baselines.
>
> In terms of generalizability, each dataset includes different temporal sampling intervals, ranging from 0.5 years (e.g., ADNI) to over 3 years (e.g., LBC). Due to this temporal discrepancy, BrainODE exhibits marginal performance decreases in both shift scenarios.
>
> These findings underscore the importance of training on large-scale longitudinal datasets with varied time intervals to enhance generalization across cohorts. Leveraging broader and more diverse training data is essential for improving robustness and capturing progression dynamics. We will include this experiment in the final version.
>
> | Experiment | Training              | Validation     | Hippo    | LV       |
> |:------------:|:-------------------------:|:-------------------:|:-------------------:|:-------------------:|
> | Exp1       | AIBL                    | ADNI + OASIS   | 0.518 (↑0.016) | 2.265 (↑0.554) |
> | Exp2       | AIBL + ADNI + OASIS     | LBC1936        | 0.522 (↑0.172) | 1.988 (↑0.203) |
>
> (Values in parentheses are the surface distance metric (mm) difference from the intra-benchmark result)
> ***
> >### (W3 & Q3) Exclusion of MCI
>
> In the clinical continuum of the AD trajectory (normal(NC) → MCI → AD), determining the severity of MCI and encoding its cognitive status values between NC and AD is highly challenging. In addition, as MCI includes multiple subtypes (e.g., EMCI, LMCI, and unspecified MCI), simply assigning an intermediate value such as 0.5 is insufficient to accurately reflect disease severity, making it difficult to clearly model and validate the BrainODE.
>
> We carefully designed BrainODE to be conditioned on continuous cognitive status values, enabling a gradual transition from NC to AD, which inherently encompasses the MCI states. Although BrainODE was trained on subjects with clearly defined cognitive status(NC or AD), we leveraged a pseudo cognitive status embedding approach on subjects whose cognition progressed from NC to AD to generate synthetic longitudinal brain shapes with intermediate cognitive status. This allowed us to train and validate BrainODE on these intermediate cognitive status. As a result, BrainODE effectively bridges the gap between NC and AD, demonstrating its capability to predict shapes on intermediate cognitive status such as MCI (Fig12).
>
> This approach not only highlights BrainODE's advantage over prior methods that rely on discrete conditioning values but also presents promising evidence that the model can flexibly predict under intermediate status such as MCI. In practical terms, this means BrainODE has the potential to be applied in real clinical settings where cognitive status is diagnosed with varying degrees of granularity by medical professionals.
>
> As the main focus of this paper is **to explore the potential of neural ODEs for modeling brain shape trajectories across a broad age range and disease conditions, we prioritized architectural design and evaluation under clearly distinguishable conditions.** In future work, by quantitatively defining the severity of MCI, our BrainODE would have stronger generalization.
> ***
> >### (W4&Q5) Granular longitudinal evaluation between 1- and 4-Shot
>
> To clarify our evaluation design, the purpose of separating 1- and 4-shot is:
>
> (1) 1-shot prediction reflects a realistic clinical use case where only a single timepoint is available
> (2) 4-shot prediction explores the model’s capability when more longitudinal data are accessible.
>
> We chose (2) 4-shot as the upper bound since the LBC dataset provides up to five longitudinal scans per subject. To clarify, the subjects used for 1-shot and 4-shot validation are not the same.
>
> To response the reviewer’s suggestion, we conducted additional experiments for 1-, 2-, 3-, and 4-shot settings in LBC1936. We observed that the prediction accuracy increased with the less standard deviations in more observations. While not the main focus of this work, we acknowledge that optimizing the aggregation of prediction from multiple timepoint is an important future direction(L705). We plan to include this result to provide a more complete picture how BrainODE scales with longitudinal depth.
> | | hippo (mm) | LV(mm)  |
> |:------:|:-----------------:|:-----------------:|
> | 1-shot | 0.46 ± 0.077   | 1.93 ± 0.270 |
> | 2-shot | 0.43 ± 0.075   | 1.84 ± 0.238 |
> | 3-shot | 0.42 ± 0.075   | 1.80 ± 0.229 |
> | 4-shot | 0.41 ± 0.075   | 1.79 ± 0.226 |
> ***
> >### (Q2)  BrainODE’s sensitivity to Cognition estimator errors
>
> To evaluate the reliability of cognition estimator, we provide the accuracy(~88%) in normal/AD classification on unseen subjects, as reported in Tab 3. Before employing the model in our BrainODE, we conducted a preliminary study on 600 subjects (1:1 normal and AD) using only the LV and hippcampus segmentation masks with the same model of Cognition estimator; it achieved over 95% classification accuracy. We interpret the probabiltiy of AD from the cognition estimator as severity of AD.
>
> About the sensitivity of BrainODE to the cognitive value, BrainODE predicts the greater atrophy for the same initial shapes under higher values, demonstrating the model’s high sensitivity to the estimator outputs (Fig3&12). This faithful reflection of cognitive status is important in our study, as clinicians can manually specify cognitive status in practical scenarios, rather than relying solely on the estimator.
>
> Nonetheless, to ensure realistic modeling without manual intervention, high precision in the cognition estimator remains important. Following your suggestion, to provide better interpretation of our BrainODE with cognition estimator, we will provide a sensitivity experiment under varying cognitive estimator accuracy.
>
> >### (Q4) ConDOR comparison
>
> ConDOR focuses on predicting future 1D brain measurements (e.g., cortical thickness as scalar) using a diffusion-based generative model, while our task requires spatially structured shape modeling in 3D mesh format, which ConDOR cannot directly accommodate. Thus, we excluded ConDOR from our baseline comparisons. Instead, we compared with diffusion-based image generative model BrLP.
>
> >### (L)
>
> We appreciate your suggestion regarding(ICLR2025) the aggregation of predictions from multiple observations. We agree with the reviewer’s comment to move the limitations discussed in appendix to the main paper to more clearly define the scope and boundaries of our study.

---

> > ### Comment · Reviewer_4KRF · 2025-08-04
> > **Response**
> >
> > Dear Authors,
> >
> > Thank you for your detailed response.
> >
> > Regarding the error bounds, I strongly encourage the inclusion of confidence intervals (CIs), as they are essential for statistically meaningful comparisons between models. Reporting only standard deviations is insufficient for this purpose and does not support inference regarding performance differences. For instance, in Table 4, the 4-shot prediction results show that Linear Extrapolation performs comparably to BrainODE, and even outperforms it for hippocampal predictions. Without CIs, it is not possible to assess whether these differences are statistically significant.
> >
> > Additionally, I would appreciate clarification regarding the inclusion of the 4-shot evaluation setting. Since the paper states that optimizing the aggregation of predictions from multiple timepoints is not a central focus of the work, it’s a bit unclear why longitudinal evaluations involving repeated measurements (as in the 4-shot setting) are included at first place. If the method is primarily designed for cross-sectional input, it would be helpful to understand the motivation behind evaluating it in longitudinal, repeated inputs of a subject. I may be missing something here so any additional explanation would be appreciated.
> >
> > Finally, I continue to believe that this work faces certain limitations that merit further attention—most notably, the handling of the MCI class, which constitutes a large portion of subjects in Alzheimer’s disease datasets. While I recognize that BrainODE is designed to model an intermediate latent state along the CN-to-AD continuum, this approach alone may not suffice for predictive inference in individuals already diagnosed with MCI. As a result, the clinical utility of the method remains constrained. I will maintain my score at this time, but I sincerely wish the authors the best of luck with their submission.

---

> > > ### Author Response · Authors · 2025-08-07
> > >
> > > ### RE: Confidence Interval
> > > Thank you for your suggestion regarding the inclusion of confidence intervals. We agree that confidence intervals provide a more statistically meaningful comparison than standard deviations alone. Since the number of subjects used in our experiments is already reported, it is feasible to compute and include 95% confidence intervals based on the provided standard deviations. For improved clarity and readability, we will incorporate both the confidence intervals and statistical significance results in the final version.
> > >
> > > If we may offer further clarification on the results from linear extrapolation, the quantitative outcomes appear to yield comparable results for modeling the longitudinal progression of brain regions. However, BrainODE demonstrates superior qualitative performance in two key aspects. First, since extrapolation does not account for 3D shape characteristics, it yields blocky and noisy outputs (Fig. 2). Second, it produces unrealistic results when extrapolating over long-time intervals (e.g., 10 years; Fig. 14,). Moreover, extrapolation is not only infeasible for predicting the future from a single observation, but it is also fundamentally incapable of capturing the complex dynamics of brain shapes across varying age ranges and disease groups.
> > >
> > > We sincerely appreciate your valuable feedback and will incorporate it into the final version of our manuscript.
> > > ***
> > > ### RE: Clarification on the 4-shot evaluation setting
> > > To clarify, BrainODE is not limited to cross-sectional input; it is also applicable in longitudinal settings. In the 4-shot evaluation, we included the subject’s four prior timepoints to train BrainODE and aimed to infer the 5th timepoint. In contrast, for the 1-shot setting, the subject was completely excluded from training.
> > >
> > > Our statement that **“optimizing the aggregation of predictions from multiple timepoints is not a central focus of the work” specifically refers to the approach used to generate the final prediction.** After training on trajectories that included the target subjects’ prior 4-shot shapes, BrainODE was used at inference time to make four separate predictions for the 5th time point, each based on a different starting point (i.e., from time points 1→5, 2→5, 3→5, and 4→5). These four predicted shapes were then simply averaged to form the final output. We did not explore more advanced average strategies such as weighted averaging or learnable fusion methods (e.g., transformer-based architectures), as this was beyond the scope of the current work.
> > >
> > > We appreciate the reviewer for pointing out the potential for confusion and will revise the manuscript to clarify this evaluation protocol. Additionally, as suggested in previous comments, we will include the 2- and 3-shot evaluations mentioned in the rebuttal above.
> > >
> > > ***
> > > ### RE: Handling of the MCI class and clinical utility
> > > We agree that addressing the MCI population is a critical challenge in Alzheimer’s disease research. As the reviewer notes, our current model focuses on learning smooth dynamics between CN and AD states in a continuous latent space. While this allows for representing intermediate states such as MCI in theory, it does not guarantee optimal performance for predictive inference specific to MCI subjects.
> > >
> > > That said, we would like to emphasize that our approach does not exclude MCI cases altogether. We acknowledge the heterogeneity inherent in MCI and consider modeling this variability an important future direction. To this end, we are currently exploring methods to explicitly incorporate heterogeneous trajectories for MCI in our follow-up studies.
> > > ***
> > > Once again, we thank the reviewer for the insightful comments and helpful suggestions.

---

### Official Review · Reviewer_VKFD · 2025-07-02

**Clarity:** 3
**Significance:** 2
**Originality:** 2
**Rating:** 4
**Confidence:** 3

**Summary:**

This paper proposes BrainODE, which resorts to a neural ODE-based framework to model continuous brain shape changes, and thereby predicting the neurodegenerative disease progression. It addresses challenges like limited data, irregular sampling, and individual variability by using a conditional architecture influenced by age and cognitive status. A pseudo-cognitive embedding further enables prediction from a single observation. In the experiments, the authors show that BrainODE outperforms existing methods in predicting future brain shapes across several different longitudinal datasets.

**Questions:**

- How would the model deal with large motion artifacts/shifts if they appear in the flow images. Would it be a potential issue if the adjacent longitudinal images do not include exactly the same viewpoint?
- How would the model be able to ensure the geometry feasibility/soundness during the transformation from the source to the target domain?
- Neural-ODE-related frameworks are typically vulnerable regarding training stability, and requires longer time and more computing resources for training. How is the model's efficiency comparing to other non-Neural-ODE models?

**Ethical Concerns:**

["NO or VERY MINOR ethics concerns only"]

**Final Justification:**

With the new results and detailed rebuttal, my concerns have been mostly resolved. I have increased my score as leaning to accept.

**Limitations:**

The authors have not sufficiently discussed their method's limitations in the submission.

**Quality:**

2

**Strengths And Weaknesses:**

Strength:
- Neural-ODEs are well-known for continuous modeling of longitudinal changes, the method is well-motivated and the design of incorporating with age and cognitive status is reasonable and sound.
- Due to the realistic challenges coming from the lack of data in clinical setups, an efficient way to address the longitudinal modeling of shape deformations could be useful for future work.
- The experiments are sufficient and clearly indicate the effectiveness of the proposed approach, evaluations on both regular and irregular time interval predictions are interesting and insightful.
- The organization and presentation is clear and easy to follow.

Weakness:
- Using Neural-ODEs to address topics regarding shape changes and longitudinal data is not novel, and there exist quite many studies on Neural-ODE/fluid-based image registration, non-rigid deformation modeling, and image sequence regression, etc ([1], [2]). However, the authors have not discuss or compare their approach with these related work.
- How would the model deal with large motion artifacts/shifts if they appear in the flow images. Would it be a potential issue if the adjacent longitudinal images do not include exactly the same viewpoint?
- How would the model be able to ensure the geometry feasibility/soundness during the transformation from the source to the target domain.
- Neural-ODE-related frameworks are typically vulnerable regarding training stability, and requires longer time and more computing resources for training. The authors have not mentioned their model's efficiency comparing to non-Neural-ODE models.

[1] Wu et al.: A Neural Ordinary Differential Equation Based Optimization Framework for Deformable Image Registration. CVPR 2022.
[2] Bai et al.: Image Sequence Regression Based on Neural Ordinary Differential Equations. MICCAI 2024.

---

> ### Author Rebuttal · Authors · 2025-07-30
>
> >### (W1) Neural-ODE Novelity and Comparison with Related Method ([1], [2])
>
> While ODE-based models have been explored in medical time-series applications, to the best of our knowledge, BrainODE is the first to model clinically important brain subregions with 3D mesh representations for longitudinal brain prediction. We position our work under the following settings, and selected baselines that can be adapted accordingly: (1) handling time-series data, (2) integrating medical priors, and (3) enabling prediction from a single observation.
>
> However, registration-based methods [1,2] are not suitable for **(3) single-shot prediction** as well as (2) unclear conditioning methods on age or cognitive status. Therefore, we exclude direct comparisons with them in the main experiment, but we agree that including a discussion of these methods in the Related Work section would enhance the clarity of our paper.
> - Regarding NODEO[1], it is primarily designed for image registration tasks by learning continuous deformation fields between moving and fixed images. While effective at interpolation between images, **NODEO lacks an explicit age-conditioning method and is not designed for extrapolation/prediction of future brain shapes**, which is central to our problem.
> - **In response to your comment, we additionally experimented with NODER [2]**, which aims to predict missing timepoints in brain image sequences. NODER requires at least two prior timepoints as input for sequence regression. As highlighted in our paper (L35 & L42), this requirement limits its practical applicability for predicting future shapes from a single observation. Moreover, it requires separate training for each individual brain image sequence, making it unsuitable for learning generalizable longitudinal dynamics.
> We conducted this experiment using the LBC1936 dataset under a 4-shot validation setting. Although NODER [2] utilizes the deformation information from earlier timepoints, its voxel-based regression approach and lack of explicit conditioning on age and disease lead to inferior predictive accuracy. This highlights the effectiveness of BrainODE’s design, which captures shape dynamics in mesh representation and explicitly conditions on age and cognitive status, enabling more accurate modeling of longitudinal shape changes. We would add this ODE-based regression method in the Appendix of the final version for completeness.
> | Method      | Hippo (mm) | LV (mm)      |
> |:-----------:|:----------------:|:------------:|
> | **NODER [2]**   |      0.898       |    4.152     |
> | **Ours**        |      **0.414**      |    **1.630**     |
>
> ***
> >### (Q1&W2) Robustness to viewpoint variation and motion artifacts
>
> The robustness across varied viewpoints is one of the key advantages of our shape-based approach. As described in Appendix A.1, we construct longitudinal subcortical brain meshes from brain MRIs by aligning them to a template mesh space. This mesh reconstruction step eliminates the influence of viewpoint variation and motion artifacts inherent in raw MRI data. To generate these aligned shapes, we:
>
> 1. Segment the target subcortical structure
> 2. Rigidly align each shape to a common template mesh M_0​ (Fig. 4) using the Iterative Closest Point algorithm.
>
> As a result, **all shapes—both intra-subject and inter-subject—are spatially aligned** before being used to train BrainODE. This alignment process ensures that BrainODE focuses **purely on anatomical deformation patterns.**
>
> ***
> >### (Q2&W3) Geometry soundness of the deformation
>
> When generating brain shapes in mesh representation, we ensure geometric plausibility using multiple mesh-related regularization loss functions (Appendix A.1). These losses promote surface smoothness and prevent skewed meshes. Additionally, we include a regularization term that encourages minimal vertex displacement from the template mesh to reconstruct individual meshes. This approach helps ensure that the shape data, used for BrainODE training, maintain geometric and anatomical soundness.
>
> On the other hand, during BrainODE training, we do not explicitly enforce geometric constraints on the predicted deformations. This design choice stems from the observation that subcortical areas are subtly changed from the original shapes as demonstrated in Fig 1, 3, and 5(c). In addition, we define and predict shape deformation in PCA spaces which inherently eliminates noisy high frequency surfaces (Appendix A.2). Thus, we do not impose hard constraints (e.g., enforcing non-zero hippocampal volume, surface smoothness), yet the model produces realistic and plausible deformations under PCA-based shape spaces. We only enforce one natural boundary condition: the mapping from time t to itself must yield the identical shape, i.e., φ(X_(t→t)) = X. This identity condition is inherently satisfied by the integral ODE of Equation 2.
>
> We acknowledge that when the model is queried with extreme out-of-distribution conditions (e.g., 120 years of age), it may produce slightly implausible shapes as discussed in Appendix C.5 and visualized in Fig 14. However, we consider such ages to be beyond practical application scenarios.
> ***
> >### (Q3&W4) Training stability and efficiency
>
> - BrainODE adopts a flow-based neural ODE framework that operates in the **deformation space, which inherently supports stable training dynamics**, as discussed in ShapeFlow. Furthermore, our **PCA-based shape representation highlights clear longitudinal deformation patterns**, which facilitate stable learning (Fig 5(a)). By leveraging this representation, BrainODE is able to preserve both global shape topology and fine-grained local variations, while predicting subtle deformations.
> - For comparison, LatentODE shows poor performance, as evidenced by producing similar shapes for an arbitrary subject (Fig 2). One possible explanation is that, despite using a mesh representation, LatentODE fails to capture longitudinal progression correctly due to training instability and the complex shape dynamics. From this perspective, we believe that our design and representation choices ensure training stability in this challenging domain.
> - Furthermore, to empirically validate the stability, we generated future hippocampal shapes from 100 randomly sampled inputs with varied cognitive statuses of 0, 0.5, and 1 (Appendix C.3). The predicted shapes were both plausible and well-reflecting the cognitive status in terms of brain atrophy, even though BrainODE was not explicitly regularized with higher cognitive scores.
> - Regarding computational efficiency, neural ODEs offer efficient gradient computation via the adjoint method as noted in L50 and NeuralODE [5]. In our implementation (Algorithm 1), we train BrainODE over all pairwise combinations of longitudinal shapes (~375MB GPU usage for training). While mini-batch training could be made more efficient with customized implementation (e.g., code of LatentODE [22]), we did not adopt this yet, but we plan to enhance its efficiency.
> - **BrainODE achieves efficient training times with 9 hours for LBC1936 with 40 epochs**, substantially more efficient than the voxel-based generative model, BrLP[20]. BrLP requires multiple training stages, including a KL-regularized VAE-GAN, diffusion model, and conditional generator (e.g., ControlNet and a generalized linear model for covariate prediction), making it significantly computationally demanding (at least one week, due to the difficulty of training a 3D VAE-GAN). We did not provide direct runtime comparisons with RNN- or LatentODE-based models since these methods failed to predict individualized shapes, making fair comparison difficult. **For transparency, we will include BrainODE’s training time in the final version.**
>
> ***
> >### (L1) Limitation on methodology
>
> In addition to the limitations described in Appendix D, one is the lack of explicit geometric constraints as discussed in this rebuttal (Q2&W3), which has a possibility to generate implausible shapes under extreme or out-of-distribution conditions.
>
> Another limitation is that BrainODE is specifically designed for brain subregions. Although BrainODE can be applied to other brain subregions, it is not directly applicable to modeling deformations of the full brain image at scale. We believe that after precisely modeling brain subregions with BrainODE, whole brains can be generated by conditional generation modules (e.g., ControlNet, VCM(2025)), leveraging the BrainODE outputs. However, we consider it beyond the scope of the present study. We will add a discussion of the above limitations to the final version of the paper.
>
> We appreciate the reviewer for raising this important point. We will include a discussion of these limitations in the main text of the final version of the paper.

---

> > ### Author Response · Authors · 2025-08-07
> >
> > Dear Reviewer VKFD,
> >
> > We would like to sincerely thank you once again for your insightful and thoughtful feedback.
> > Your comments were invaluable in clarifying key aspects of our work and motivated us to conduct additional experiments.   We believe these improvements address your concerns and further strengthen the overall contribution of the paper.
> >
> > If you have further thoughts or additional feedback, we would be grateful to hear them. Your continued engagement would be highly appreciated.
> >
> > Thank you again for your time and consideration.
> >
> > Sincerely,
> > The authors

---

> > > ### Comment · Reviewer_VKFD · 2025-08-08
> > >
> > > Thank you for the new results and detailed rebuttal, my concerns have been mostly resolved. I have increased my score accordingly.

---

> > > > ### Author Response · Authors · 2025-08-09
> > > >
> > > > We sincerely appreciate your comments. We will carefully revise our manuscript in light of your insightful feedback, and we are grateful for your guidance throughout the review process.

---

### Official Review · Reviewer_ViBW · 2025-07-02

**Clarity:** 3
**Significance:** 2
**Originality:** 2
**Rating:** 4
**Confidence:** 3

**Summary:**

The authors provide motivation from their application (modelling brain deformations due to ageing / dementia using MRI data) for a 3D shape deformation model that can incorporate known covariates (here they chose age and cognitive variables). Neural ODEs are a reasonable approach for this. They compare their method to off-the-shelf methods (RNN-types) and an existing method in the field (BrLP) and demonstrate superior performance under certain metrics.

**Questions:**

L17 - knowledge of the hippocampus's role in AD is not recent, it has been reported many years ago (at least 1991 in the Braak & Braak paper)

L27 - where do the numbers 160x224x160 with 1mm^3 resolution come from?

L34 - what do you mean by "medical priors"?

L40 - what do you mean by "not invertible" in this context?

L58 - term "condition-injectivity" is italicised as if it has a specific meaning, but it's never defined, just assumed. What does it mean?

L144 - how does age and cognitive status incorporate prior knowledge on brain atrophy and ventricle enlargement? Do you just mean that you expect them to change monotonically wrt. age and cognitive variables?

L148 - need ablation study for effect of doing both forward and backward solvers (i.e., only forward or only backward). It is not clear why you need to include both

L152 - what is "approximated bijectivity"?

L152 - I don't quite follow the inline equation; are you saying the transformation matrix \Phi is orthogonal (inverse equals transpose)?

Equation 1 - what does f_\theta(\Lamba_t, c_t, t) equal? The neural ODE? And how is the conditioning on c_t actually implemented? It's not clear from the text.

Equation 1 - does this equation have boundary conditions?

Algorithm 1 - if I read this correctly, you sequentially solve the ODE over all timepoints, then all timepoints - 1, then all time-points - 2, and so on, with the initial conditions incrementing correspondingly. What's the justification for this? Why not just solve once for the initial timepoint and write the loss with respect to the predictions at each followup data point? The way it's written, you're essentially solving the parts of the same ODE multiple times, instead of a single continuous mapping across all time points; is there not redundancy in the solution here?

Algorithm 1 - why N-2 going forward (i.e., requiring you need at least 2 timepoints) and N-1 going backward?

Algorithm 1 - what package and arguments did you use for "ODEintegral"? If it's in the Supplementary Material it needs to be moved to the main text, as it has strong implications for solver stability, runtime, etc.

L154 - it may be my limitation, but I can't understand why this is called "pseudo-cognitive status embedding". From what I understand, you're just adding training points by interpolating between timepoints using your ODE solver. What does this have to do with "pseudo-cognitive status"?

**Ethical Concerns:**

["NO or VERY MINOR ethics concerns only"]

**Final Justification:**

I am satisfied with the authors' responses and have increased my score to "Borderline Accept", but not full "Accept", due to concerns about the overall novelty of the work.

**Limitations:**

Please see my comments above concerning limitations.

No comment was made by the authors on societal impact.

**Paper Formatting Concerns:**

None.

**Quality:**

3

**Strengths And Weaknesses:**

Strengths
- Neural ODEs are a reasonable approach for the task
- Their method can handle irregularly sampled data, which are common in medical analysis

Weaknesses
- The novelty is unclear; ODE-based methods for this task are common
- Most of the baseline models are insufficient for the task (RNN-types are not designed for this problem)
- There are technical flaws in the methodology that need addressing (please see my comments below)

---

> ### Author Rebuttal · Authors · 2025-07-28
>
> ### (W1)
> BrainODE is the first neural ODE-based framework for modeling longitudinal 3D shape trajectories of the brain in individual level and disease group at scale. Here, **we identify a novel problem: the precise modeling of longitudinal shape deformation in clinically important brain subregions**, extending beyond prior works focused on whole-brain image prediction.
>
> We highlight the novelty through three main contributions:
>
> 1. **Bridging neural ODE-based 3D shape modeling with longitudinal neurodegenerative disease**:
>
> We propose a dedicated pipeline consisting of (i) mesh reconstruction for longitudinal shape representation, (ii) PCA-based parameterization, and (iii) a unified deformation-based ODE framework across disease groups. While neural ODEs have shown promise in continuous-time modeling, they have not been applied to predict longitudinal brain subregion shapes. To fill this gap, we convert brain images to fine-grained meshes and represent them in a compact PCA space. Then, we introduce BrainODE with continuous conditioning, pseudo-status embedding, and iterative shape sampling, to ensure both modeling stability and disease-continuum plausibility. Our model achieves precise subject-level trajectory prediction with low computational cost (~375MB for training), while maintaining fidelity at both the individual and group levels.
>
> 2. **First large-scale application of longitudinal brain shapes across age and multi-site datasets**:
>
> To the best of our knowledge, BrainODE is the first framework to apply Neural ODEs to model longitudinal brain shapes across a wide 30 years age span and multi-site datasets. Through large-scale experiments, we demonstrate that BrainODE effectively captures brain morphological patterns and aging dynamics across both normal and AD trajectories. Furthermore, we identify fundamental limitations in existing generative image-based models, which fail to precisely capture subregion-level deformation, and we offer straightforward adaptations of ODE methods to address this limitation.
>
> 3. **Simultaneous identification and resolution of core challenges in longitudinal shape modeling**:
>
> We define and simultaneously address key challenges unique to longitudinal brain shape modeling. These include data scarcity, irregular sampling intervals, the integration of medical priors, and the ability to predict only from a single observation. Even if beating SOTA performance is not a main goal, by defining these considerations upfront (L33-45), BrainODE provides a foundational blueprint for this research area.
> ***
> ### (W2)
> - Given the limited research on future brain shape prediction, we selected time-aware architectures as baselines. These include RNN- and ODE-based approaches, motivated by prior works such as LatentODE [22], which specifically addresses irregularly sampled time series—similar to the longitudinal brain data used in our study.
> - In addition, we included BrLP [20], an image-based conditional generative model that also predicts future brain structures conditioned on age and cognitive status. Given its similar objective, BrLP serves as a particularly relevant baseline, and we provide a detailed comparison in Appendix C.4.
> ***
> ### (W3)
> We thank the reviewer for pointing out areas of ambiguity. Below, we provide clear and detailed responses to each of your comments.
> ***
> >### L17
> We acknowledge that the hippocampus has long been recognized as critical in AD. The intention to cite a recent study [15] is not to define clinical meaning of hippocampus, but rather to highlight that **hippocampal shape alone can be a strong indicator for AD diagnosis**. It supports the importance of precise modeling of the clinically important subregions.
>
> >### L27
> The specified image size is a typical brain volume that can **accommodate most adult brains. The 1mm³ voxel spacing is a common setting** used in many representative brain imaging studies (BrainLDM(2022), SynthSeg (2023)).
> Here, we would like to emphasize that the hippocampus occupies only ~0.2% of the whole brain with subtle deformation over time. Since 1mm voxel representations are often insufficient to capture such localized changes, we adopt a mesh-based strategy.
>
> >### L34
> The term **“medical priors” refers to prior knowledge from clinical studies about brain shape changes** according to age and cognitive status (L38, L268). To support this, we offer this volume analysis on real datasets in Appendix B.2.
> We observed that hippocampal volume tends to decrease, while that of LV increases, as age progresses or as cognitive status declines (e.g., CN → AD). These monotonic trends highlight that brain shape dynamics are significantly influenced by these variables. Accordingly, **BrainODE is conditioned on age and cognitive status** as well as on the current brain shape to effectively predict within these deformation patterns.
>
> >### L40
> The term “not invertible” was used to convey that brain structural changes associated with neurodegenerative diseases are typically **not reversible** (i.e., once significant atrophy occurs, the brain does not return to a healthy state). To avoid confusion, we will revise the term “not invertible” to “not reversible”.\
> The intention behind L40 was to emphasize our use of a **continuous cognitive status variable**, not relying solely on discrete diagnostic labels (e.g., normal = 0, AD = 1) as done in prior works [20]. This continuous label allows BrainODE to flexibly reflect gradual cognitive decline.
>
> >### L58
> **“Condition-injectivity” refers to BrainODE’s ability to produce different future shapes depending on the given conditions** (age and cognitive status), even from the same current shape. A subject with an AD trajectory produces more severe atrophy than one with a normal (Fig 3 & Appendix C.3). We will clarify this in the final version.
>
> >### L144
> As addressed in our response to L34, hippocampal volume decreases while that of LV increases over aging with an accelerated pattern (Appendix B.2&Fig7). In addition, AD subjects demonstrate more pronounced deformations compared to normal (Fig8).
> Importantly, **we expect that BrainODE captures these progression patterns, without the need for explicit constraints** enforcing greater atrophy at higher ages or cognitive impairment levels. Fig 12 confirms that the model successfully learns and reflects these trends.
>
> >### L148
> **Since only the forward solver is used for prediction**, the backward training does not improve future shape prediction. Thus, we did not include an ablation experiment for the backward process on predictive performance.
> **However, this backward solver plays a key role in our pseudo-cognitive shape sampling: it is used to predict past shapes from brain shapes with AD trajectory**, enabling denser longitudinal supervision during BrainODE training (Section 3.2, L155).
>
> >### L152
> - Ideally, the forward and backward processes are bijective, i.e., φ_ji(φ_ij(Λ)) = Λ.
> However, we use the term **“approximated” bijectivity as shape predictions rely on numerical integration**, which cannot strictly guarantee perfect invertibility (the lemma for the deformation bijectivity is discussed in ShapeFlow).
> The inline equation is intended to express this bijectivity: if a past shape at time t is reconstructed from a predicted future shape at t + δt, it should recover the original input X using the backward solver.
> - Here, **φ refers to the neural ODE-based shape mapping defined in Equ 2 & L146**.
> We clarify that φ is not a transformation matrix and nor assumed to be orthogonal. Rather, it represents a learned **mapping from time t to another time t′.** We will clarify the bijectivity and φ in the final version.
>
> >### Equation 1
> - Yes, f_θ is BrainODE (Fig6), and details are in Appendix A.3.
> - As demonstrated in multiple visual examples (Fig1, 3, 5(c)), BrainODE learns subtle shape changes in longitudinal datasets. Therefore, we do **not impose hard constraints**(e.g., enforcing that hippocampal volume remains strictly non-zero in future predictions). One boundary condition is that mapping from **timepoint t to itself should return the identical shape, i.e., φ(X_(t→t)) = X**, which is inherently satisfied by the integral ODE process.
>
> >### Algorithm 1
> - We use all available pairwise timepoints (source&prediction) to maximize the available training data.
> - **There is no redundancy in the for loop.** In the forward process, the initial source shape’s timepoint i ranges from 0 to N−2, and for each i, all subsequent timepoints from i+1 to N−1 are predicted. The final timepoint (N−1) is excluded as an initial point since there are no future timepoints. Conversely, the initial timepoint ranges from N−1 down to 1 in the backward process.
> (cf) We acknowledge an error in Alg1- line 2, where the range should be from 0 to N−1 (not 1 to N). We will correct this, and we appreciate your careful attention.
> - The function odeIntegral is based on the `odeint` function from the torchdiffeq library, used to implement Equ2. While the details are currently in L514 (Appendix A.4), we agree that moving them to the main text would improve clarity.
>
> >### L154
> **Pseudo-cognitive status embedding is a data augmentation strategy to train BrainODE with continuous cognitive status** Our dataset includes only subjects diagnosed as normal (CN) or AD. Importantly, there are several subjects (labeled as CONV in Tab5) that exhibit disease progression from CN to AD. Considering the disease progression continuum (CN→congnition impairment→AD), we leverage the CONV subjects to synthesize intermediate brain shapes using BrainODE, and assign corresponding intermediate cognitive status. Since these values **are not directly observed but inferred from the progression, we refer to them as pseudo cognitive status.** It enables BrainODE to learn a smoother mapping along the cognitive decline continuum.
> ***
> We sincerely appreciate your thoughtful comments, which will help enhance the clarity of our work.

---

> > ### Comment · Reviewer_ViBW · 2025-08-05
> >
> > I thank the reviewers for their thoughtful answers to my questions. I have increased my score accordingly.

---

> > > ### Author Response · Authors · 2025-08-09
> > >
> > > We are truly grateful for your insightful and constructive feedback. Your comments have guided us in improving the clarity of our work, and we deeply appreciate your guidance throughout the review process.

---

### Note · Authors · 2025-08-13

We thank the reviewers and AC for their insightful comments, which helped us clarify and strengthen our work. This is the final summary for our main contributions and key clarifications.
- Challenges addressed
We identify three challenges in longitudinal brain shape prediction and address them at scale:
(1) predicting future shapes on arbitrary timepoint by training the longitudinal dynamics from datasets with irregular time intervals,
(2) incorporating medical prior knowledge and
(3) predicting future shapes from a single baseline observation.

- Proposed method
We propose BrainODE, a framework that models time-continuous deformations of brain shapes conditioned on continuous age and cognitive status using neural ODE architecture. Shapes are reconstructed by deforming a template mesh (ensuring point correspondence), projected into PCA space for a compact yet accurate representation. This comprehensive pipeline achieves optimal performance in predicting future brain shapes.

- Treatment of intermediate cognitive status
Although MCI subjects were excluded in this study, we would like to say this is not a fundamental limitation, as BrainODE is designed to model longitudinal brain shape changes rather than to serve as a disease prediction model for the entire Alzheimer’s disease (AD) spectrum. Furthermore, BrainODE learns the progression from normal to AD as a continuous trajectory, enabling plausible predictions for intermediate cognitive states through continuous variable conditioning, pseudo cognitive status embedding, and the corresponding shape sampling strategy (Fig12). This suggests BrainODE can be extended to MCI, and we plan to expand it by developing the principled method to determine MCI cognitive status in future work.
Given that longitudinal brain shape modeling is in early stages, it is crucial to identify the inherent challenges and address them systematically. BrainODE meets these objectives and can provide valuable insights for advancing toward more general and comprehensive studies.

- Rebuttal process
We carefully addressed all comments and resolved most concerns. In addition, we conducted further experiments in: (1) gradual-shot shape prediction, (2) comparison with another baseline model, (3) validation across benchmark datasets, and (4) evaluation with additional shape alignment metrics. These results further confirm the superiority of our approach. We will incorporate our responses into the final manuscript accordingly.

---

### Decision · Program_Chairs · 2025-09-17

**Decision:**

Accept (poster)

**Comment:**

(a) Scientific claims and findings: The paper presents BrainODE, a neural‑ODE framework for longitudinal 3D brain shape modeling at the subregion level (notably hippocampus and lateral ventricles). Shapes are reconstructed as meshes and embedded in a PCA shape space; their temporal dynamics are governed by an ODE conditioned on age and cognitive status. A pseudo‑cognitive embedding augments training to interpolate along a CN→AD continuum and supports one‑shot (single‑visit) forecasting under irregular sampling.

Across four longitudinal MRI cohorts, BrainODE predicts anatomically plausible future shapes and outperforms RNN/LSTM, Latent ODE, flow‑based/registration and image‑generative baselines (including BrLP) under both one‑shot and multi‑shot settings. The rebuttal adds cross‑dataset generalization experiments, 2‑/3‑shot analyses, and statistical reporting (CIs/significance to be included in the final version).

(b) Strengths:

1. Problem importance & clinical relevance: Modeling subregion shape trajectories is central to aging and AD progression; focusing on anatomically meaningful meshes improves interpretability.
2. Methodological fit: Neural ODEs naturally capture continuous‑time trajectories and irregular intervals; conditioning on age/cognition encodes well‑established medical priors (atrophy/enlargement trends).
3. One‑shot forecasting: The pseudo‑cognitive augmentation enables prediction from a single observation, a realistic clinical scenario.
4. Engineering choices that matter: Working in mesh‑PCA space reduces noise and cost while preserving fine geometric detail; training remains lightweight relative to voxel‑generative pipelines.
5. Empirical breadth: Multiple cohorts, one‑shot and multi‑shot regimes, added cross‑dataset validation, and qualitative analyses (trajectory smoothness, plausibility). Rebuttal commits to 95% CIs/significance tests, subset reporting by dataset, and additional metrics (e.g., Dice).
6. Clarity improvements: Rebuttal resolves many terminology/formulation questions (conditioning, bijectivity, solver choice, algorithmic loops) and moves limitations into the main paper.

(c) Weaknesses / Missing elements
1. Scope of anatomy: Current evaluation is on hippocampus and ventricles; extension to cortical surfaces and more complex regions remains future work.
2. MCI coverage: MCI (a clinically pivotal intermediate stage) is not modeled explicitly; the pseudo‑cognitive continuum is promising but not a substitute for dedicated MCI inference.
3. Geometric guarantees: Predicted deformations are not constrained by hard geometric regularizers; plausibility is aided by mesh‑space priors and data preprocessing, but formal guarantees are absent.
4. Baseline coverage / statistics: Initial draft lacked CIs/significance and some domain‑specific metrics; rebuttal addresses this (to be added) but final inclusion is important.
5. Reliance on cognition estimator: Performance may depend on estimator accuracy; robustness and sensitivity analyses are only partially explored (promised in revision).

(d) Reasons for acceptance

Timely, high‑impact problem with clear translational value: precise subregional shape forecasting for aging and AD progression.
Strong integration of ideas: clinically informed conditioning + continuous‑time dynamics + mesh‑PCA representation + pseudo‑cognitive augmentation—delivering one‑shot forecasting under irregular sampling.
Compelling empirical evidence: consistent improvements over diverse baselines, plus new cross‑dataset and multi‑shot analyses; qualitative results demonstrate trajectory smoothness and anatomical plausibility.
Practicality and efficiency: orders‑of‑magnitude lighter than voxel‑generative pipelines while targeting a finer (subregional) prediction task.

Thorough, constructive rebuttal resolving conceptual and methodological questions and committing to statistical rigor and clearer presentation.

(e) Rebuttal‑period discussion and how it informed the decision

Key issues raised (anon ids R1…R4) and author responses:

Motivation, terminology, and formulation (R1): Asked to clarify “medical priors,” (ir)reversibility, “condition‑injectivity,” “approximate bijectivity,” and the exact ODE/conditioning mechanics; queried Algorithm‑1 loops and solver choice.
Response: Definitions clarified; “not invertible” → “not reversible”; φ explained as a learned ODE mapping (not an orthogonal matrix); boundary condition specified; torchdiffeq/odeint disclosed; loop‑range bug fixed; justification for pairwise training provided.

Baselines, novelty, and related work (R2/R4): Requested deeper comparisons to neural‑ODE registration, flow‑matching, diffusion, and ConDOR; questioned 4‑shot rationale; suggested stronger metrics/visualizations.
Response: Added NODER comparison (favoring BrainODE); argued registration/ConDOR are mismatched to mesh‑shape forecasting; committed to discussing ImageFlowNet/flow‑matching/diffusion; clarified 4‑shot protocol and added 2‑/3‑shot results; will include Dice and enhanced trajectory plots.

Statistics and reporting (R2): Called for confidence intervals and significance testing; noted cases where extrapolation looked comparable.
Response: Will add 95% CIs + significance; argued qualitative/long‑horizon and one‑shot scenarios favor BrainODE over extrapolation.

Generalization, robustness, and geometry (R3): Asked for cross‑dataset testing, sensitivity to cognition estimator, handling motion/viewpoints, geometric soundness, and extension beyond subcortical regions; raised MCI omission.
Response: New cross‑dataset experiments; mesh alignment/regularization for viewpoint/motion; PCA‑space acts as a soft geometric prior; cognition estimator accuracy reported with planned sensitivity analysis; acknowledged cortical extension as future work; justified MCI handling via pseudo‑continuum but recognized it as a limitation to be moved to the main text.

Weighting in the final decision:
The rebuttal substantively addressed the highest‑impact concerns (formulation clarity, evaluation breadth, statistical rigor commitments, and cross‑dataset evidence). Remaining limitations (explicit MCI modeling, cortical extension, formal geometric guarantees) are reasonable for future work and do not undermine the contribution’s core value. With three score increases and full‑panel acceptance, the paper clears the bar for acceptance.